# Growth differentiation factor 1-induced tumour plasticity provides a therapeutic window for immunotherapy in hepatocellular carcinoma

Wei Cheng [1,2], Hao-Long Li[1,2], Shao-Yan Xi[3], Xiao-Feng Zhang[1,2], Yun Zhu[4], Le Xing[1,2], Yan-Xuan Mo[1,2], Mei-Mei Li[1,2], Fan-En Kong [1,2], Wen-Jie Zhu[1,2], Xiao-Gang Chen[1,2], Hui-Qing Cui[1,2], Zhi-Ming Cao[1,2], Yuan-Feng Gong[1,2], Yun-Qiang Tang[1,2], Yan Zhang[4], Xin-Yuan Guan [5], Ning-Fang Ma [1,2] & Ming Liu [1,2✉]

Tumour lineage plasticity is an emerging hallmark of aggressive tumours. Tumour cells usually hijack developmental signalling pathways to gain cellular plasticity and evade therapeutic targeting. In the present study, the secreted protein growth and differentiation factor 1 (GDF1) is found to be closely associated with poor tumour differentiation. Overexpression of GDF1 suppresses cell proliferation but strongly enhances tumour dissemination and metastasis. Ectopic expression of GDF1 can induce the dedifferentiation of hepatocellular carcinoma (HCC) cells into their ancestral lineages and reactivate a broad panel of cancer testis antigens (CTAs), which further stimulate the immunogenicity of HCC cells to immune-based therapies. Mechanistic studies reveal that GDF1 functions through the Activin receptor-like kinase 7 (ALK7)-Mothers against decapentaplegic homolog 2/3 (SMAD2/3) signalling cascade and suppresses the epigenetic regulator Lysine specific demethylase 1 (LSD1) to boost CTA expression. GDF1-induced tumour lineage plasticity might be an Achilles heel for HCC immunotherapy. Inhibition of LSD1 based on GDF1 biomarker pre-screening might widen the therapeutic window for immune checkpoint inhibitors in the clinic.

[1] Affiliated Cancer Hospital and Institute of Guangzhou Medical University, Guangzhou 510095, China. [2] Guangzhou Municipal and Guangdong Provincial Key Laboratory of Protein Modification and Degradation, School of Basic Medical Science, Guangzhou Medical University, Guangzhou 511436, China. [3] State Key Laboratory of Oncology in Southern China, Collaborative Innovation Center for Cancer Medicine, Sun Yat-sen University Cancer Center, Guangzhou, China. [4] Department of Pediatric Surgery, Guangzhou Institute of Pediatrics, Guangzhou Women and Children's Medical Center, Guangzhou Medical University, Guangzhou, China. [5] Department of Clinical Oncology, State Key Laboratory for Liver Research, The University of Hong Kong, Hong Kong, China. ✉email: liuming@gzhmu.edu.cn

Hepatocellular carcinoma (HCC) is one of the most common human malignancies with a poor prognosis, and it ranks as the fourth leading cause of cancer mortality worldwide[1]. Although surgical resection and liver transplantation are current options for early-stage HCC treatment, most patients are diagnosed at a late stage and miss the opportunity for curative therapy. Currently, several multikinase inhibitors, including sorafenib, regorafenib, and lenvatinib, have been approved by the US Food and Drug Administration (FDA) as first-line or second-line treatments for unresectable advanced HCC. However, the benefit of therapy is very limited, with a prolonged median overall survival time of less than 3 months[2–4]. The emergence of immune-checkpoint inhibitors, such as monoclonal antibodies directed against programmed cell death protein 1 (PD-1), has recently shown promising results in HCC[5]. However, only a small fraction of patients responds to immune-based therapies. More biomarkers indicative of drug response and therapeutic strategies to maximise the therapeutic effects are urgently needed.

Clinical high-grade tumours usually show phenotypic resemblance to their ancestral cells and hijack developmental signalling pathways to gain lineage plasticity[6]. Cancer is a dynamic disease that is highly plastic and continues to evolve during malignant progression and therapeutic treatment. Increasing evidence indicates the existence of a stem cell hierarchy in the tumour bulk, which harbours distinct molecular signatures and cellular identities[7,8]. Tumour cells undergo phenotypic switching between different cellular lineages, resulting in tumour heterogeneity, which substantially contributes to therapeutic resistance[9,10]. Cellular plasticity has been observed in both normal liver development and HCC malignant transformation[11,12]. Cancer cells of hepatic origins and bile duct epithelial origins can switch identities upon activation of certain developmental signalling pathways and hamper effective therapeutic targeting of HCC[13,14]. Our recent finding also confirmed the existence of a developmental hierarchy in HCC with different oncofetal regulators[15]. Many factors can affect the effectiveness of immune checkpoint inhibitor therapy, including tumour mutation burden (TMB), programmed cell death-ligand 1 (PD-L1) expression, T-cell infiltration, and human leucocyte antigen (HLA) diversity[16]. Recent studies have indicated that cancer stemness is not only a fundamental process in cancer progression but also may provide a mechanistic link to the immune response[17]. However, due to the complexity of cancer stem cell properties, the association between the stemness index and immune signatures varies across multiple tumour types[18]. The mechanisms underlying the effects of cancer stemness and tumour plasticity on the immune therapeutic response are still unclear.

In the present study, GDF1, which belongs to the transforming growth factor-β (TGF-β) superfamily, is found to be highly expressed in poorly differentiated high-grade HCC tumours. Overexpression of GDF1 suppresses cell proliferation but significantly enhances tumour invasion and metastasis both in vitro and in vivo. The expression of GDF1 is silenced in most mature tissues but activated in embryonic development. Forced expression of GDF1 or addition of GDF1-recombinant protein can induce tumour plasticity in HCC cells, which will then exhibit biomarkers of liver progenitors. We also find that a series of cancer testis antigens (CTAs) are significantly activated in GDF1-overexpressing tumour cells, and this process might be mediated through inhibition of the epigenetic modulator LSD1. Although GDF1-induced cell plasticity enables the malignant transformation of HCC, reactivation of CTAs might be an Achilles heel for immunotherapy in this subtype of tumour. Our in vivo mouse model demonstrates that GDF1-overexpressing tumours show enhanced cytotoxic T-cell infiltration and higher sensitivity to anti-PD1 therapy. Here, we show that GDF1-induced tumour plasticity can sensitise HCC cells to immune checkpoint inhibitors. Epigenetic drug LSD1 inhibitor, which can activate CTAs, may boost the immunogenicity of tumour cells and further enhance the therapeutic effects of immune-checkpoint inhibitors in the clinic.

## Results

**GDF1 was highly expressed in high-grade poorly differentiated HCCs.** To identify the gene expression profile associated with the tumour microenvironment of poorly differentiated tumours, a PCR array containing detection probes for genes encoding 375 secreting chemokines or cytokines was performed using 3 pooled HCC tissues with poor differentiation and their paired nontumor liver tissues (Supplementary Fig. 1a and Supplementary Data 1). The TGF-β superfamily was enriched with the most differentially expressed genes. Among them, GDF1 had the highest fold change (Supplementary Fig. 1b). The relative mRNA expression of GDF1 was measured in the HKU cohort, which included 83 HCC patients. Significant upregulation of GDF1 was found in HCC tumour tissues compared with their paired normal counterparts (Fig. 1a). The upregulation of GDF1 was further confirmed in The Cancer Genome Atlas (TCGA) database LIHC project, which included 373 HCC patients and 50 paratumor liver tissues (Fig. 1b). Both western blot and immunohistochemical staining (IHC) were performed to confirm the overexpression of GDF1 at the protein level in representative paired HCC samples (Fig. 1c, d). Since GDF1 was isolated from high-grade tumours, we further measured the expression of GDF1 in subgroups of patients with different tumour grades. A significant progressive increase in GDF1 from low-grade tumours to high-grade tumours was found in both the HKU cohort and TCGA cohort (Fig. 1e, f). IHC staining also confirmed the high expression of GDF1 in poorly differentiated tumours compared with moderate- and well-differentiated tumours (Fig. 1g).

**Clinical significance of GDF1 expression in HCC patients.** To precisely determine the expression of GDF1 at the protein level and examine the association of GDF1 staining with the clinicopathological features of HCC patients, IHC staining for GDF1 was performed in a tissue microarray (TMA) containing 196 liver tumour tissues from HCC patients. The relative expression of GDF1 was defined according to the scoring system (Supplementary Fig. 1c). High staining of GDF1 (score > =2) was detected in 127 out of 196 (64.8%) HCC patient samples examined. Kaplan–Meier analysis indicated that the high expression of GDF1 was significantly correlated with poor overall survival (log-rank test, $P < 0.0001$, median overall survival, GDF1 high: 37 months; GDF1 low: undefined, Fig. 1h) and disease-free survival of HCC patients (log-rank test, $P = 0.0018$, median disease-free survival, GDF1 high: 12 months; GDF1 low: 127 months, Fig. 1i). A clinicopathological association study indicated that high expression of GDF1 was significantly associated with adjacent invasion, clinical stage, tumour relapse, and poor differentiation (two-sided $\chi2$ test, Supplementary Table 1). Furthermore, univariate and multivariate Cox regression analyses also showed that the high expression of GDF1 was an independent prognostic factor in HCC ($P = 0.001$, Supplementary Table 2). Considering the negative staining in paratumour liver tissues, the expression of GDF1 was further examined in a panel of normal tissues from the Genotype-Tissue Expression (GTEx) database. GDF1 expression remained low or absent in most of the normal tissues, except for the genital organs and brain (Fig. 1j). This finding indicated that GDF1 is silenced in most normal organs, and reactivation of GDF1 in the tumours might provide an ideal therapeutic target for HCC.

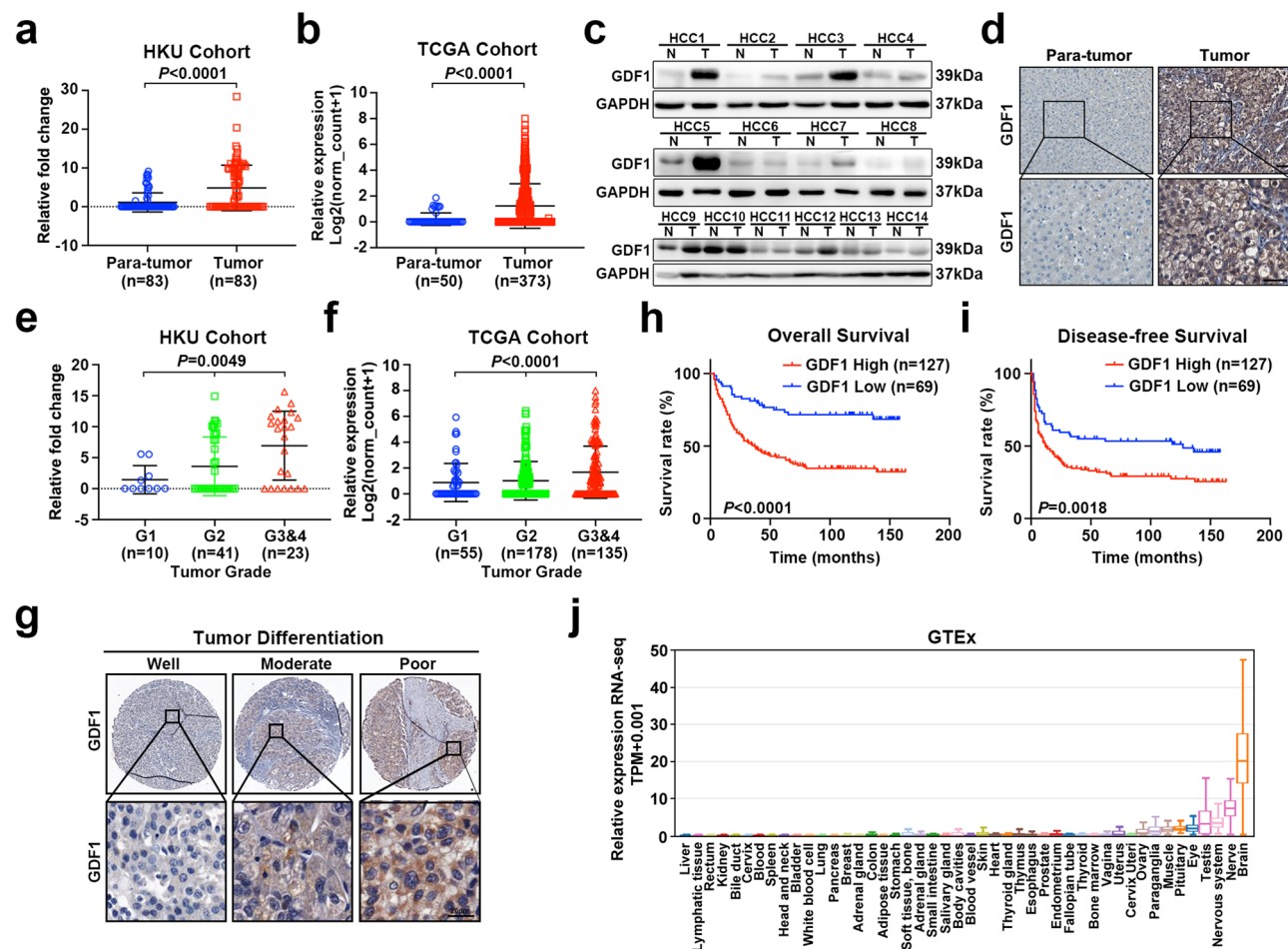

**Fig. 1 The expression and clinical significance of GDF1 in HCC. a** The relative mRNA expression of GDF1 was detected by qPCR in the HKU cohort, which included 83 HCC tissues and their paired nontumor liver tissues (two-tailed paired-sample *t*-test, data are presented as mean values ± SD). **b** The relative mRNA expression of GDF1 was measured by qPCR in the TCGA cohort, which included 373 HCC patients and 50 paratumor liver tissues (two-tailed independent Student's *t*-test, data are presented as mean values ± SD). **c** The relative expression of GDF1 at the protein level was measured by western blot in representative HCC tissues and their paired nontumor liver tissues (*n* = 3 independent experiments). **d** IHC staining of GDF1 in representative HCC tissue and paired nontumor liver tissue (*n* = 3 independent experiments). **e** The relative mRNA expression of GDF1 was measured by qPCR in subgroups of patients with different tumour grades in the HKU cohort (one-way ANOVA, data are presented as mean values ± SD) **f** and in the TCGA cohort (one-way ANOVA, data are presented as mean values ± SD). **g** IHC staining of GDF1 in representative poor-, moderate-, and well-differentiated HCC tumours in a tissue microarray (TMA) containing 196 liver tumour tissues. **h, i** Kaplan–Meier analysis indicated that the high expression of GDF1 was significantly correlated with poor overall survival (**h**) and disease-free survival (**i**) of HCC patients (log-rank test). **j** The expression of GDF1 was examined in a panel of normal tissues from the GTEx database (17,382 RNA-seq across 54 tissue sites and two cell lines). Boxes represent quartiles, centre lines denote 50th percentile, and whiskers extend to most extreme values. Scale bars represent 20 µm. Source data are provided as a Source Data file.

**Overexpression of GDF1 suppressed HCC cell proliferation but strongly induced tumour metastasis**. To test the functional roles of GDF1 in HCC, the relative mRNA expression of GDF1 was screened in a series of HCC cell lines (Supplementary Fig. 1d). PLC-8024 and Huh7 cells with relatively low GDF1 expression were stably transfected with full-length GDF1, and overexpression was confirmed at the protein level by western blot (Fig. 2a). Overexpression of GDF1 significantly suppressed the proliferation and colony-formation ability of PLC-8024 cells (Fig. 2b, c). Considering the dual role of the TGF-β family in both suppressing cell growth and promoting tumour metastasis, we further examined whether GDF1 affects HCC tumour metastasis. In vitro functional assays showed that overexpression of GDF1 significantly enhanced migration and invasion abilities in both PLC-8024 and Huh7 cells (Fig. 2d, e). Similar results were observed in wild-type PLC-8024 cells cocultured with PLC-8024-GDF1 cells or expressed recombinant GDF1 protein (Supplementary Fig. 1e, f). Immune-deficient mice with intrasplenic

injection of HCC cells also showed a significant increase in metastatic liver and lung tumour nodules in the GDF1-overexpressing group compared with the control group (Fig. 2f). Pathological characterisation of the metastatic tumours was further confirmed by H&E staining, and the positive staining of GDF1 was confirmed by IHC (Fig. 2g).

**Overexpression of GDF1 induced HCC tumour-lineage plasticity**. Considering that GDF1 is silenced in mature hepatocytes but reactivated in HCC, we inferred that GDF1 might be associated with liver development. Foetal mouse livers at different developmental stages were collected, and the relative expression of GDF1 and several developmental biomarkers was measured by qPCR. As shown in Fig. 3a, GDF1 exhibited a peak in expression at embryonic day 16.5 (E16.5), which mirrored the expression of several liver progenitor markers, such as Keratin19 (KRT19), Keratin7 (KRT7), SRY-box transcription factor 9 (SOX9) and Alpha fetoprotein (AFP). The activation of GDF1 was also confirmed by IHC staining in foetal

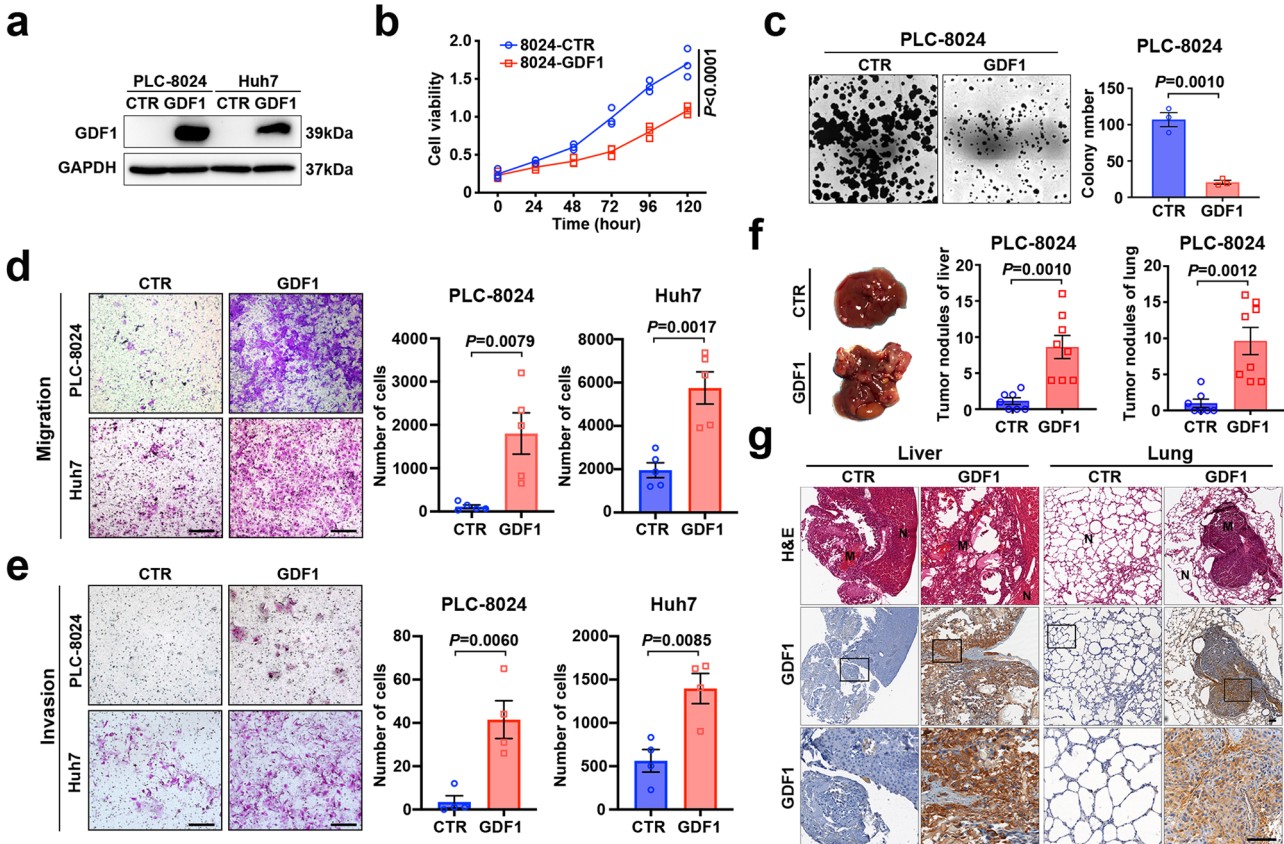

**Fig. 2 GDF1 suppressed HCC cell proliferation but strongly induced tumour metastasis. a** PLC-8024 and Huh7 cells were stably transfected with full-length GDF1, and overexpression was confirmed at the protein level by western blot ($n = 3$ independent experiments). **b** A cell-viability assay was performed to evaluate the cell proliferation rate in PLC-8024-CTR cells and PLC-8024-GDF1 cells (two-way ANOVA test, data are presented as mean values ± SEM, $n = 3$ independent experiments). **c** A colony-formation assay was performed to evaluate the colony formation ability of PLC-8024-CTR cells and PLC-8024-GDF1 cells (two-tailed independent Student's $t$ test, data are presented as mean values ± SEM, $n = 3$ independent experiments). **d, e** Cell migration (**d**) and invasion (**e**) abilities were measured in PLC-8024 cells and Huh7 cells stably transfected with or without GDF1 (two-tailed independent Student's $t$-test, data are presented as mean values ± SEM, **d**: $n = 5$ independent experiments, **e**: $n = 4$ independent experiments). **f** A total of PLC-8024-CTR cells and PLC-8024-GDF1 cells were intrasplenically injected into immune-deficient BALB/c nude mice. Metastatic liver and lung tumour nodules were calculated 6 weeks later (two-tailed independent Student's $t$-test, data are presented as mean values ± SEM, PLC-8024-CTR: $n = 7$, PLC-8024-GDF1: $n = 8$). **g** H&E staining and IHC staining of GDF1 were performed in metastatic liver and lung tumour nodules (PLC-8024-CTR: $n = 7$, PLC-8024-GDF1: $n = 8$). Scale bars represent 200 μm. Source data are provided as a Source Data file.

mice liver (Supplementary Fig. 2a). Considering that the retro-differentiation occurs during partial hepatectomy, the staining of GDF1 was further examined in regenerating mice liver after partial hepatectomy. As shown in Supplementary Fig. 2b, positive staining of GDF1 was found in the mice liver 3–5 days after surgery. These findings indicated that GDF1 might be important for liver development and regeneration. To test whether GDF1 affects normal hepatocyte proliferation, primary mice hepatocytes were cultured in an organoid model, and subjected to lentivirus-mediated transfection of GDF1 or control vector. As shown in Supplementary Fig. 2c, overexpression of GDF1 significantly inhibited primary hepatocytes proliferation reflected by Ki67 staining. This was in accordance with the findings in HCC cell lines, that GDF1 could suppress cell proliferation. Sphere-formation assays also showed that overexpression of GDF1 significantly enhanced the self-renewal ability of both PLC-8024 and Huh7 cells (Fig. 3b). To further test whether GDF1 can induce HCC cells toward their ancestral lineage, representative hepatic markers Albumin (ALB), Alcohol dehydrogenase 1 (ADH1), Arginase 1 (ARG1), Transthyretin (TTR), and liver progenitor markers AFP, Epithelial Cell Adhesion Molecule (EPCAM), KRT19, SOX9, and KRT7 were examined in PLC-8024 and Huh7 cells transfected with GDF1 or a control vector. Overexpression of

GDF1 significantly induced the expression of liver progenitor markers but inhibited the expression of terminal differentiated hepatic markers (Fig. 3c and Supplementary Fig. 2d). A significant positive correlation of GDF1 with liver progenitor markers and a negative correlation with hepatic markers were found in the TCGA–LIHC cohort[19] (Supplementary Fig. 2e). Furthermore, coculture of PLC-8024-GDF1 cells with wild-type PLC-8024 cells or addition of recombinant GDF1 protein also led to the significant upregulation of liver progenitor markers and inhibition of hepatic markers accordingly (Fig. 3d, e). The dedifferentiation of HCC cells induced by GDF1 was measured at the protein level by western blot in both PLC-8024 cells and Huh7 cells (Fig. 3f). IHC staining of xenograft tumours further confirmed tumour plasticity and lineage reversion in HCC cells with GDF1 overexpression (Fig. 3g). In addition, GDF1 costained with liver progenitor markers in poorly differentiated HCC tissues (Fig. 3h). These findings indicated that overexpression of GDF1 might enhance HCC tumour plasticity and change the fate of HCC cells toward the ancestral lineage.

**GDF1 functioned mainly through the ALK7–SMAD2/3 signalling cascade.** The fundamental mechanism of TGF-β superfamily

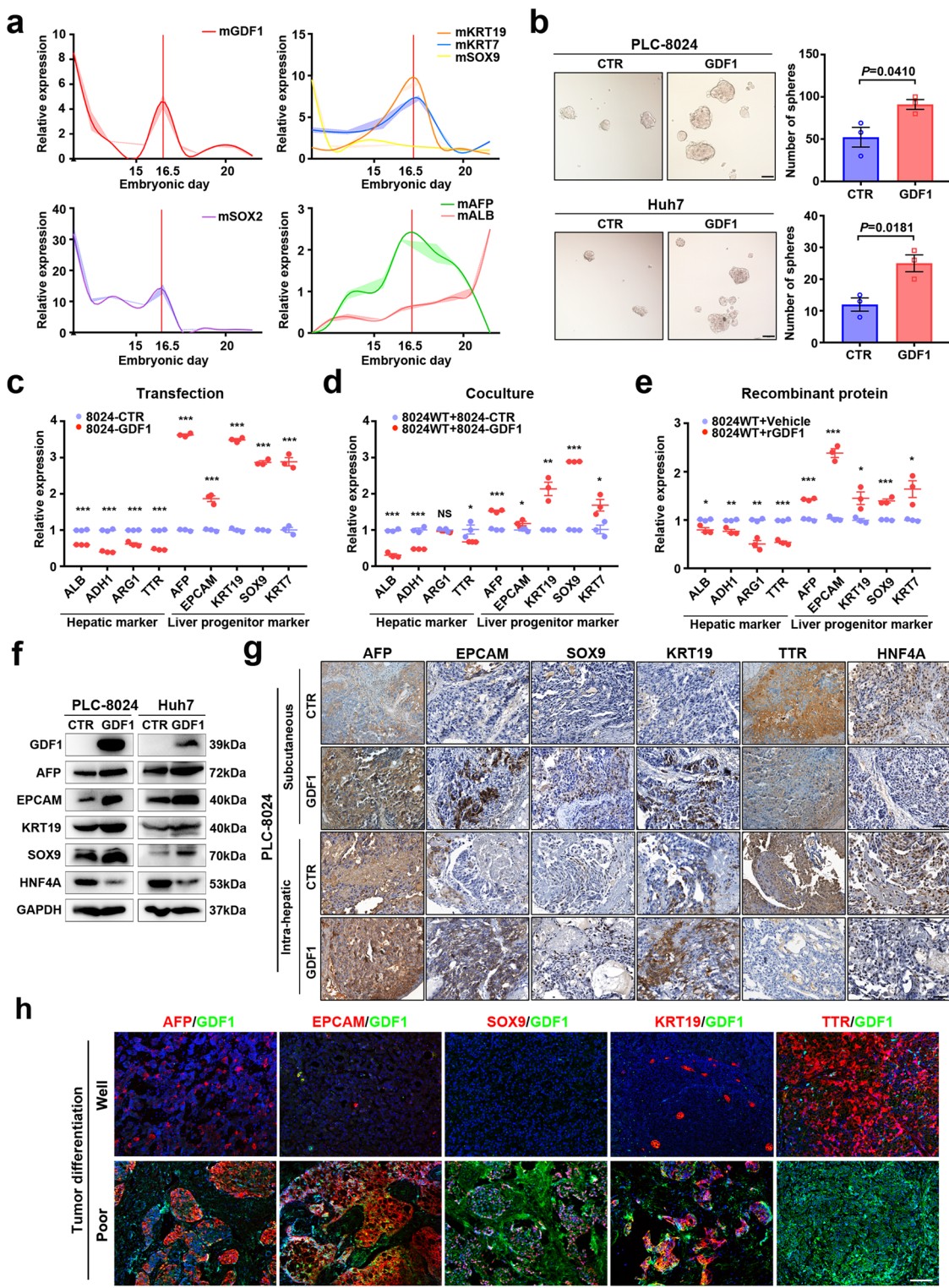

activation involves ligand binding to a heterodimer of serine/threonine–kinase receptors. Upon phosphorylation, the receptors activate downstream SMAD cascades, which further transduce signals into the nucleus and initiate the transcriptional network. Different ligands use different receptors and compositions of their downstream SMAD complexes to transduce specific signalling cascades, which constitutes the complicated TGF-β superfamily network during development and cancer progression[20]. GDF1 was reported to share the receptor ALK4/7 with Nodal and GDF3, which further activate downstream SMAD3 signalling. To

test which TGF-β superfamily receptor was involved in GDF1-induced HCC tumour lineage plasticity, PLC-8024 cells stably transfected with or without GDF1 were transduced with lentivirus-mediated shRNAs specifically targeting different ALKs (ALK4/5/7), respectively (Fig. 4a). We found that the enhanced cell migration induced by GDF1 overexpression was totally abolished by knocking down ALK7, but not ALK4 or ALK5 (Fig. 4b and Supplementary Fig. 3a). Similar result was observed in the sphere-formation assay in PLC-8024 cells transfected with or without GDF1 (Fig. 4c and Supplementary Fig. 3b). These

**Fig. 3 Overexpression of GDF1 induced HCC tumour-lineage plasticity. a** Foetal mouse livers at different developmental stages were collected, and the relative expression of GDF1 and several developmental biomarkers was measured by qPCR. Shaded areas show the standard error of the means, $n = 3$ independent experiments. **b** A sphere-formation assay was performed in PLC-8024 cells and Huh7 cells stably transfected with or without GDF1 (two-tailed independent Student's t-test, data are presented as mean values ± SEM, $n = 3$ independent experiments). Scale bars represent 100 μm. **c–e** Representative hepatic markers (ALB, ADH1, ARG1, and TTR) and liver progenitor markers (AFP, EPCAM, KRT19, SOX9, and KRT7) were examined by qPCR in PLC-8024 cells stably transfected with or without GDF1 (**c**), coculture of wild-type PLC-8024 cells with PLC-8024-GDF1 cells (**d**), or addition of recombinant GDF1 protein for 15 days (**e**) (*$P < 0.05$, **$P < 0.01$, ***$P < 0.001$, NS, not significant, two-tailed independent Student's t test, data are presented as mean values ± SD, $n = 3$ independent experiments). **f, g** The retrodifferentiation of HCC cells induced by GDF1 was examined at the protein level by western blot in both PLC-8024 cells and Huh7 cells (**f**) ($n = 3$ independent experiments), and by IHC staining of subcutaneous and orthotopic liver tumours (**g**) ($n = 5$ independent experiments). Scale bars represent 100 μm. **h** Immunofluorescent costaining of GDF1 (green) with liver progenitor markers and hepatic markers (red) was examined in HCC clinical tissues. Scale bars represent 100 μm ($n = 3$ independent experiments). rGDF1, recombinant human GDF1 protein. Source data are provided as a Source Data file.

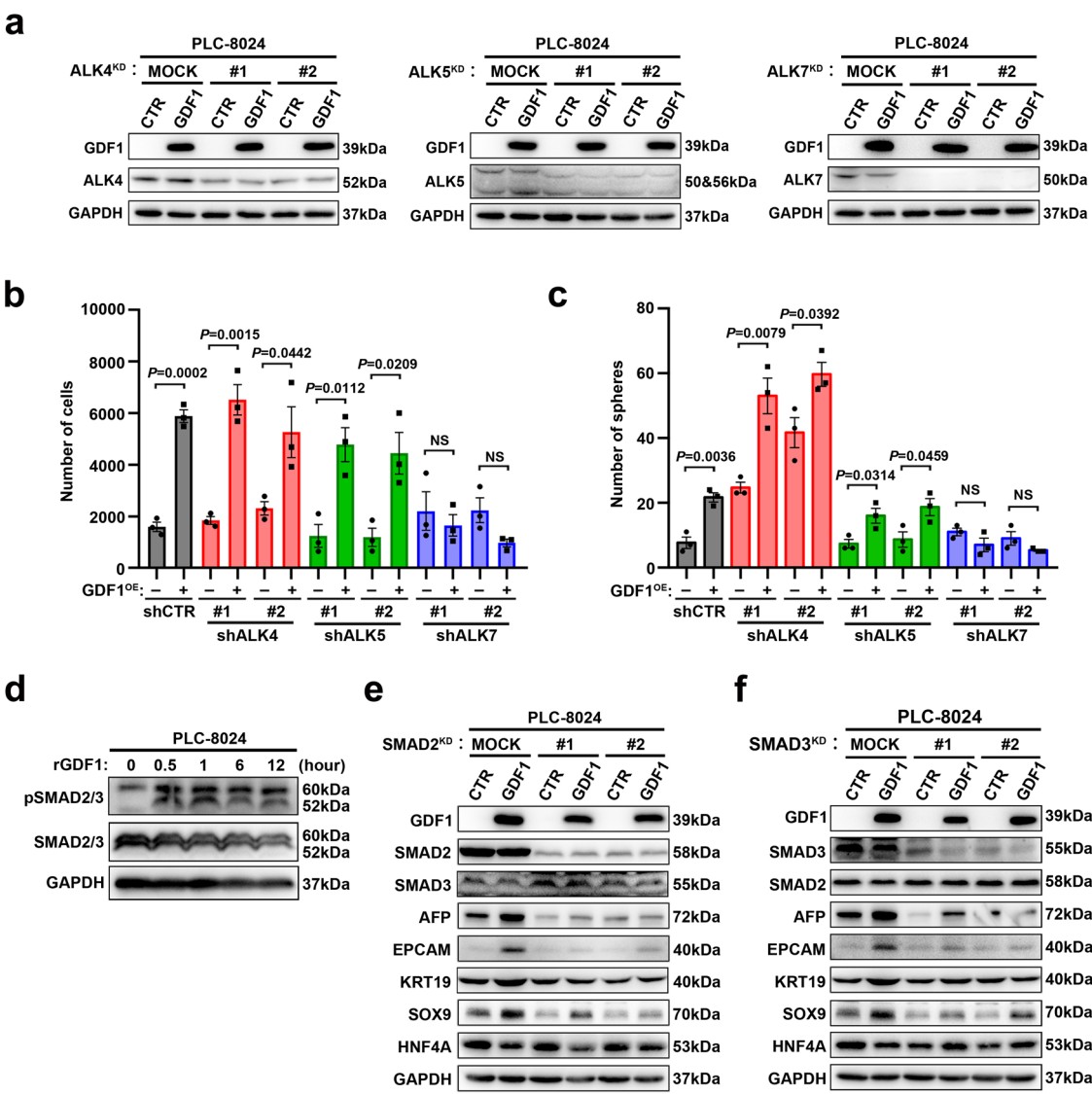

**Fig. 4 GDF1 acted mainly through the ALK7–SMAD2/3 signalling cascade. a** PLC-8024 cells transfected with or without GDF1 were transduced with lentivirus-mediated shRNAs specifically targeting ALK4, ALK5, and ALK7, respectively. Knockdown efficiency was confirmed at protein level by western blot ($n = 3$ independent experiments). **b** Cell migration assay was performed in subgroups targeting different ALKs (NS, not significant, two-tailed independent Student's t test, data are presented as mean values ± SEM, $n = 3$ independent experiments). **c** Sphere-formation assay was performed in subgroups targeting different ALKs (NS, not significant, two-tailed independent Student's t-test, data are presented as mean values ± SEM, $n = 3$ independent experiments). **d** Recombinant GDF1 protein (50 ng/mL) was added in PLC-8024 cells, the phosphorylated and total SMAD2/3 were detected by western blot at indicated time points ($n = 3$ independent experiments). **e, f** The liver progenitor markers and hepatic differentiation markers were examined by western blot in PLC-8024-CTR and PLC-8024-GDF1 cells transduced with lentivirus-mediated shRNAs specifically targeting SMAD2 (**e**) or SMAD3 (**f**) ($n = 3$ independent experiments). Source data are provided as a Source Data file.

results indicated that the ALK7 receptor, but not ALK4 or ALK5, was mainly responsible for the oncogenic function of GDF1 in HCC. As SMAD2/3 is the major executive SMAD downstream of ALK7, we further tested whether GDF1 could activate SMAD2/3 in HCC. The results showed that treatment with recombinant GDF1 protein at 50 ng/ml increased the phosphorylation of SMAD2/3 in a time-dependent manner (Fig. 4d). In addition, shRNAs specifically targeting SMAD2 or SMAD3 both abolished the upregulation of liver progenitor markers induced by GDF1 overexpression (Fig. 4e, f). These findings indicated that GDF1-induced HCC tumour plasticity and malignant phenotypes were mainly through the ALK7–SMAD2/3 signalling cascade.

**GDF1 activated a broad panel of cancer-testis antigens (CTAs) in HCC.** To further investigate the transcriptome regulated by GDF1, RNA-seq was performed using PLC-8024 cells transfected with GDF1 or control vectors. Interestingly, we found that many differentially expressed genes were enriched in the gene sets of cancer-testis antigens, which include GAGE family members (Fig. 5a and Supplementary Fig. 4a). The upregulation of G-antigen (GAGE) family genes and other CTAs was validated by qPCR in PLC-8024 cells and other HCC cells transfected with GDF1 or control vectors (Fig. 5b and Supplementary Fig. 4b, c). Cocultivation of PLC-8024 cells with PLC-8024-GDF1 cells or the addition of recombinant GDF1 protein also significantly upregulated CTAs (Fig. 5c, d). Since the GAGE family members share great similarity, the expression of the representative GAGE family member GAGE12E was further examined by western blot and IHC staining. Upregulation of GAGE12E was detected in both PLC-8024 cells and Huh7 cells transfected with GDF1 compared with control groups (Supplementary Fig. 4d). IHC staining of GAGE12E in subcutaneous or intrahepatic xenograft tumours formed by PLC-8024-GDF1 cells or PLC-8024-CTR cells further confirmed the strong upregulation of representative CTAs upon GDF1 overexpression (Fig. 5e). In addition to the GAGE family genes, the regulatory role of GDF1 on other representative CTAs was examined in PLC-8024 cells. We found that overexpression of GDF1 can also activate the expression of CTAs including melanoma antigen (MAGE) and lymphocyte antigen 6 (LY6) family members (Supplementary Fig. 4e). Although no significant correlations were found between GDF1 and GAGE family members, which might be due to no expression of GAGEs in certain proportion of HCC patients, significant positive correlation was found between GDF1 and MAGE or LY6 family members (Supplementary Fig. 4f). These findings indicated that GDF1-induced tumour-lineage plasticity also enhanced the expression of a broad panel of CTAs in HCC.

**GDF1 suppressed the epigenetic regulator LSD1 to enhance CTA expression.** CTAs are silenced in most normal organs, except for genital organs such as the testis. Epigenetic mechanisms are the major cause of CTA silencing[21]. To examine which epigenetic regulator is mainly responsible for GDF1-induced CTA expression, a panel of epigenetic modulators was screened by qPCR in HCC cells transfected with or without GDF1 (Supplementary Fig. 5a). Lysine-specific histone demethylase 1 (LSD1, also named KDM1A) was significantly downregulated by GDF1 in both PLC-8024 cells and Huh7 cells (Fig. 5f). In addition, gene set enrichment analysis (GSEA) indicated in the downstream genes regulated by GDF1, LSD1 targets were enriched (Supplementary Fig. 5b). The inhibition of LSD1 at the protein level by GDF1 was also confirmed by western blot analysis (Supplementary Fig. 5c). To further determine whether inhibition of LSD1 could boost the expression of CTAs in HCC cells, both PLC-8024 cells and Huh7 cells were treated with an LSD1-specific inhibitor,

and the panel of CTAs regulated by GDF1 was detected by qPCR. As shown in Fig. 5g, a small molecular inhibitor of LSD1 significantly induced the expression of a panel of CTAs in HCC cells. The activation of representative CTA GAGE12E by LSD1 inhibitor in PLC-8024 cells was also confirmed by western blot in a dose-dependent manner (Supplementary Fig. 5d). Interestingly, the activation of GAGE12E induced by LSD1 inhibition was greatly boosted in the presence of GDF1 overexpression (Fig. 5h). To further examine whether GDF1 is required for GAGE12E activation induced by LSD1 inhibition, sgRNA-guided GDF1-knockout cell line was established in Hep3B cells. As shown in Supplementary Fig. 5e, although LSD1 inhibitor can still activate GAGE12E expression in the absence of GDF1, the extent of activation was diminished compared to that in the wild-type Hep3B cells. These findings indicated that GDF1 expression might be a potential indicator for LSD1 inhibitors in activation of CTAs in HCC.

**SMAD2/3 bound to the promoter region of LSD1 and inhibited its expression in a GDF1-dependent manner.** To further investigate how GDF1 suppresses LSD1 expression, the binding of SMAD2/3 on the promoter region of LSD1 was predicted and validated by ChIP-PCR and luciferase assay. Data from the Encyclopaedia of DNA Elements database (ENCODE, https://www.encodeproject.org/) indicated the occupation of SMAD2/3 on the promoter region of LSD1 in several cell lines (Supplementary Fig. 5f). ChIP-PCR assay further confirmed the binding of SMAD2/3 on LSD1 promoter under stimulation with either recombinant GDF1 or TGF-β1 (Fig. 5i). Surprisingly, the luciferase-reporter assay revealed that the transcription of LSD1 was suppressed by SMAD2/3 under stimulation with GDF1, but activated under stimulation with TGF-β1 (Fig. 5j). qPCR results further confirmed that TGF-β1 could suppress the expression of CTAs and activate the expression of LSD1 (Supplementary Fig. 5g). The opposite role of GDF1 and TGF-β1 in regulating LSD1 expression indicated that SMAD2/3 might regulate downstream signalling in a case-dependent manner. Considering that SMAD2/3 usually functions through recruiting different cofactors to regulate target-gene expression, it is possible that stimulation with GDF1 or TGF-β1 may induce different cofactors that bind with SMAD2/3 for gene regulation. To further confirm whether GDF1 is required for SMAD2/3-induced LSD1 inhibition, ectopic expression of SMAD3 was induced in GDF1-knockout cell lines. As shown in the results, overexpression of SMAD3 failed to inhibit LSD1 expression and activate GAGE12E when GDF1 is depleted (Fig. 5k). In addition, specific inhibition of SMAD2 or SMAD3 abolished the regulation of GDF1 on both LSD1 and GAGE12E expression (Supplementary Fig. 5h, i). These findings indicated that the components of the GDF1/SMAD2/3/LSD1 axis äre mutually dependent, and might function only in specified tumour microenvironments.

**GDF1-induced tumour-lineage plasticity might sensitise HCC patients to anti-PD1 therapy.** As CTAs are not expressed in most antigen-presenting cells from normal tissues, their reactivation in tumour tissues was proposed to be an ideal target for immune-based therapies, such as T cell receptor-T cell (TCR-T) and cancer vaccines[22]. Many molecular and cellular processes determinants have been linked to the responses to immune checkpoint inhibitors. Among them, tumour mutational burden (TMB) was proposed to be one of the most important predictors of anti-PD1 therapy[23]. Considering that the neoantigens generated by gene mutation or fusion may greatly stimulate T cell responses[24], reactivation of CTAs might also enhance tumour immunogenicity and predict responses to checkpoint inhibitors[25].

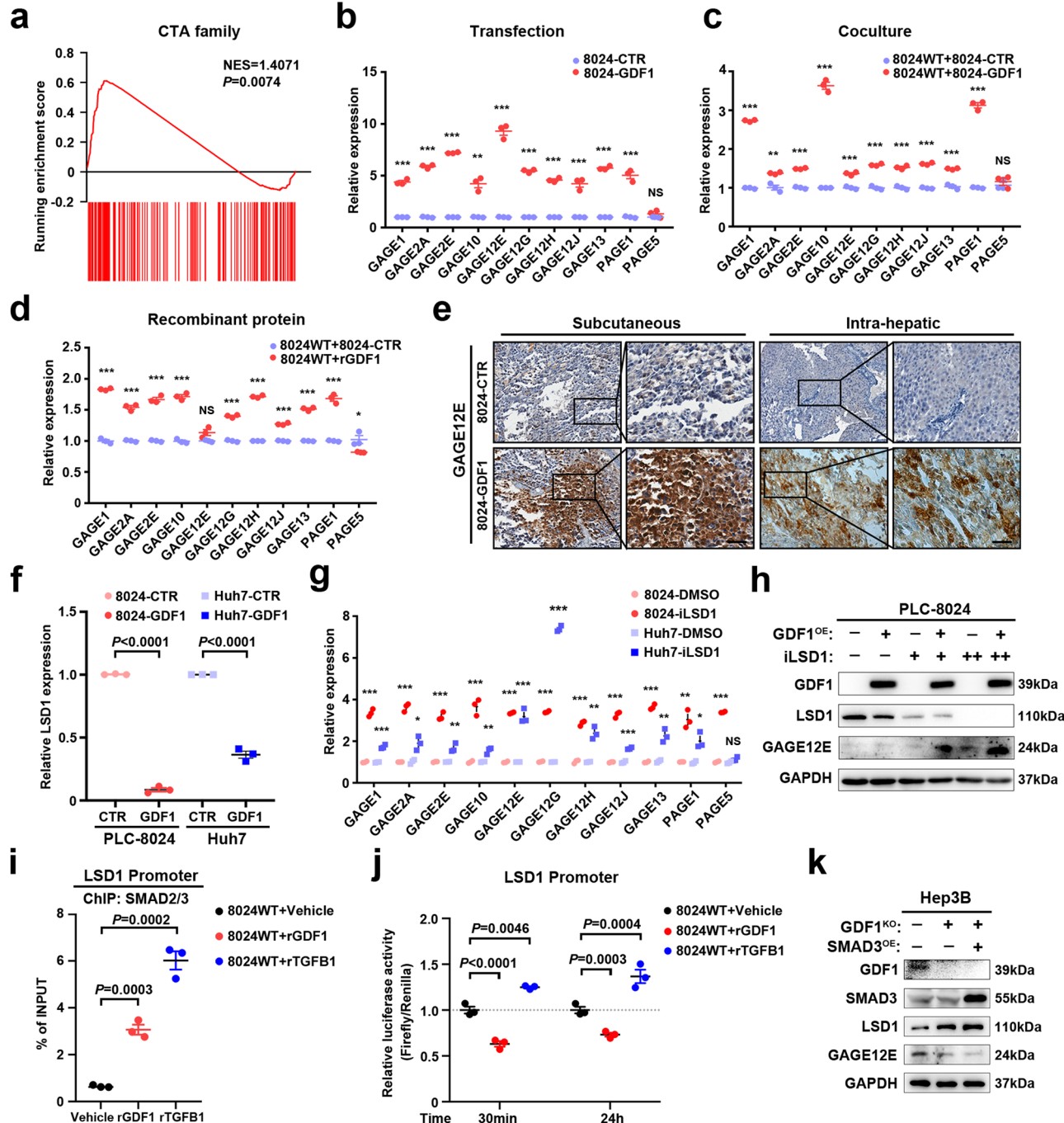

A recent study stratified HCC patients into immune-specific groups and a nonimmune group according to a panel of inflammatory markers. The immune-specific group was further divided into two subgroups: those with an activated immune response and those with an exhausted immune response[26]. To test the effect of GDF1 in the HCC immune response, HCC patients from the TCGA–LIHC project were divided into three subgroups (the activated immune group, exhausted immune group, and remaining group) according to the specified classifiers (Supplementary Fig. 6). The expression of GDF1 and the cytotoxic T-cell surface marker CD8, which reflects tumour-infiltrated cytotoxic T cells, was found to be significantly higher in the activated immune group than in the exhausted immune group and the remaining nonimmune group (Fig. 6a, left). In contrast, levels of the immune suppressors TGF-β1 and PDL1 were

significantly higher in the exhausted immune group than in the activated immune group (Fig. 6a, right). These results suggested that, unlike the traditional immunosuppressive TGF-β1, GDF1 might lead to a favoured immune microenvironment for HCC. To further test whether GDF1 affects the immune response to anti-PD1 therapy in HCC, the mouse HCC cell line Hepa1-6 was stably transfected with mouse full-length GDF1 or a control vector. Cells were intrasplenically injected into immune-competent C57BL/6 mice. The cells were labelled with luciferase, and in vivo imaging was used to track tumour growth and dissemination in the animals (Supplementary Fig. 7a). Mice bearing Hepa1-6 tumours transfected with mGDF1 or control vector were treated with anti-PD1 antibodies. As shown in Fig. 6b, mGDF1-transfected tumours were significantly more sensitive to anti-PD1 therapy than control tumours

**Fig. 5 GDF1 activated a broad panel of cancer-testis antigens (CTAs) by suppressing the epigenetic regulator LSD1 in HCC. a** RNA-seq was performed in PLC-8024-CTR and PLC-8024-GDF1 cells. Gene set enrichment analysis (GSEA) was performed in PLC-8024 cells transfected with or without GDF1 to examine the enrichment in gene sets of cancer-testis antigens. NES, normalised enrichment score. **b** The relative expression of representative CTAs was detected by qPCR in PLC-8024-CTR and PLC-8024-GDF1 cells (*$P < 0.05$, **$P < 0.01$, ***$P < 0.001$, NS, not significant, two-tailed independent Student's $t$ test, data are presented as mean values ± SD, $n = 3$ independent experiments). **c** The relative expression of representative CTAs was detected by qPCR in PLC-8024 cells cocultured with PLC-8024-CTR and PLC-8024-GDF1 cells (**$P < 0.01$, ***$P < 0.001$, NS, not significant, two-tailed independent Student's $t$-test, data are presented as mean values ± SD, $n = 3$ independent experiments). **d** The relative expression of representative CTAs was detected by qPCR in PLC-8024 cells treated with rGDF1 at 50 ng/mL or vehicle control for 15 days (*$P < 0.05$, ***$P < 0.001$, NS not significant, two-tailed independent Student's $t$ test, data are presented as mean values ± SD, $n = 3$ independent experiments). **e** IHC staining of GAGE12E in subcutaneous and intrahepatic tumours formed by PLC-8024-CTR and PLC-8024-GDF1 cells. Scale bars represent 50 µm, $n = 3$ independent experiments. **f** The relative expression of representative LSD1 was detected by qPCR in PLC-8024 and Huh7 cells transfected with or without GDF1 (two-tailed independent Student's $t$-test, data are presented as mean values ± SD, $n = 3$ independent experiments). **g** PLC-8024 and Huh7 cells were treated with the LSD1 inhibitor GSK-LSD1 at 5 µM for 72 h. The relative expression of representative CTAs was detected by qPCR (*$P < 0.05$, **$P < 0.01$, ***$P < 0.001$, NS, not significant, two-tailed independent Student's $t$ test, data are presented as mean values ± SEM, $n = 3$ independent experiments). **h** PLC-8024-CTR and PLC-8024-GDF1 cells were treated with LSD1 inhibitor at 5 µM or 10 µM for 72 h. The expression of GAGE12E and LSD1 at the protein level was examined by western blot. iLSD1, GSK-LSD1. +, 5 µM. ++,10 µM ($n = 3$ independent experiments). **i** PLC-8024 cells were treated with rGDF1 at 50 ng/mL or TGF-β1 at 10 ng/mL or vehicle control for 30 min. The enrichment of SMAD2/3 on LSD1 promoter under stimulation with either recombinant GDF1 or TGF-β1 was conducted by ChIP-PCR assay (two-tailed independent Student's $t$-test, data are presented as mean values ± SEM, $n = 3$ independent experiments). **j** Promoter activities of LSD1 in PLC-8024 cells treated with rGDF1 at 50 ng/mL or TGF-β1 at 10 ng/mL or vehicle control for indicated time points were detected by dual-luciferase assay (two-tailed independent Student's $t$-test, data are presented as mean values ± SEM, $n = 3$ independent experiments). iLSD1, GSK-LSD1. rGDF1, recombinant human GDF1 protein. rTGFB1, recombinant human TGF-β1 protein. **k** Ectopic expression of SMAD3 was induced in GDF1-knockout cell lines ($n = 3$ independent experiments). The expression of GAGE12E and LSD1 at the protein level was examined by western blot. Source data are provided as a Source Data file.

(Supplementary Fig. 7b). In addition, although prometastatic GDF1 is strongly associated with a poor prognosis in C57BL/6 mice, anti-PD1 therapy dramatically reversed malignant progression and significantly prolonged the overall survival of C57BL/6 mice bearing GDF1-overexpressing tumours (Fig. 6c). The metastatic lung nodules were further characterised with H&E staining (Supplementary Fig. 7c). Mice bearing Hepa1–6 tumours transfected with mGDF1 also showed a significant decrease in lung metastasis after anti-PD1 therapy compared with what was observed in the control subgroups of mice (Supplementary Fig. 7d). IHC staining showed that the tumour-infiltrating CD8+/ Granzyme B (GZMB) + T lymphocytes were greatly increased in mGDF1-transfected tumours after anti-PD1 therapy (Fig. 6d). To further confirm our findings in clinical HCC patients, we analysed the expression of GDF1 and CD8+-infiltrating T lymphocytes in consecutive TMA slides. Interestingly, we found that the presence of high CD8+ T-lymphocyte infiltration significantly prolonged the OS and DFS of HCC patients with high GDF1 expression (Fig. 6e, f). However, CD8+ T lymphocyte infiltration did not significantly affect prognosis in patients with low GDF1 expression. In addition, a significant correlation between GDF1 and CD8 staining was also found in HCC patients (Fig. 6g and Supplementary Fig. 7e, f).

**Combination of LSD1 inhibitor and anti-PD1 antibody might provide therapeutic strategy for HCC patients**. CRISPR–Cas9-mediated GDF1 knock out Hepa1-6 cells were established and intrasplenically injected into C57BL/6 mice to evaluate its metastatic potential (Supplementary Fig. 8a). As shown in the results, depletion of GDF1 abolished the metastatic ability of Hepa1–6 cells when high concentration of cells was injected ($3.5 \times 10^6$ cells/mice) (Supplementary Fig. 8b–d). This indicated that GDF1 might be required for the metastasis of HCC. Considering that inhibition of LSD1 can strongly induce CTA expression, we further tested the in vivo therapeutic potential of combining LSD1 inhibitor with anti-PD1 antibody. As shown in the results, combination of LSD1 inhibitor with anti-PD1 antibody showed significant superior therapeutic advantages compared with single-treatment groups in both suppressing tumour growth and metastasis, and prolonging overall survival (Fig. 6h–j and

Supplementary Fig. 8e–g). IHC staining also showed that the tumour infiltrating CD8+/GZMB+ cytotoxic T lymphocytes were greatly increased in the combination group than in the single-treatment groups (Supplementary Fig. 8h).

Taken together, our findings indicated that GDF1 is reactivated in high-grade HCCs and promotes the strong metastatic ability of this cancer. Overexpression of GDF1 induced tumour dedifferentiation toward the ancestral lineage and upregulated a broad panel of CTAs, which also provided a therapeutic window for immune checkpoint inhibitors. Inhibition of the epigenetic modulator LSD1 in GDF1-positive HCC tumours strongly boosted the expression of CTAs, which might further sensitise HCC patients to immune-based therapies (Supplementary Fig. 8i).

**Discussion**

Tumour-lineage plasticity has emerged as one of the most important mechanisms of therapeutic resistance in recent years[9,10]. Tumour development resembles developmental processes, with both intrinsic factors such as transcriptional programmes and extrinsic factors such as the tumour microenvironment, which may affect tumour-cell fate and lineage plasticity[27,28]. The common feature between tumour-lineage plasticity and developmental processes is the activation of potential cancer stem cells in the tumour and re-expression of progenitor cell markers, which usually remain low or are no longer expressed in normal terminally differentiated cells[29]. In addition to the transcriptional machinery of the tumour, temporal and spatial dynamic expression of environmental factors also plays critical roles in tumour cell fate determination[30]. To better understand the environmental factors important for HCC tumour lineage plasticity and to search for potential prognostic biomarkers, a secreting chemokine PCR array was performed to profile HCC tumours with poor pathological differentiation. We found that TGF-β superfamily members were enriched in the most differentially expressed genes, and GDF1 was the top-ranked gene.

TGF-β is important in a broad spectrum of cellular processes and plays critical roles in malignant cancer transformation[31]. Alterations in the TGF-β signalling pathway have been frequently

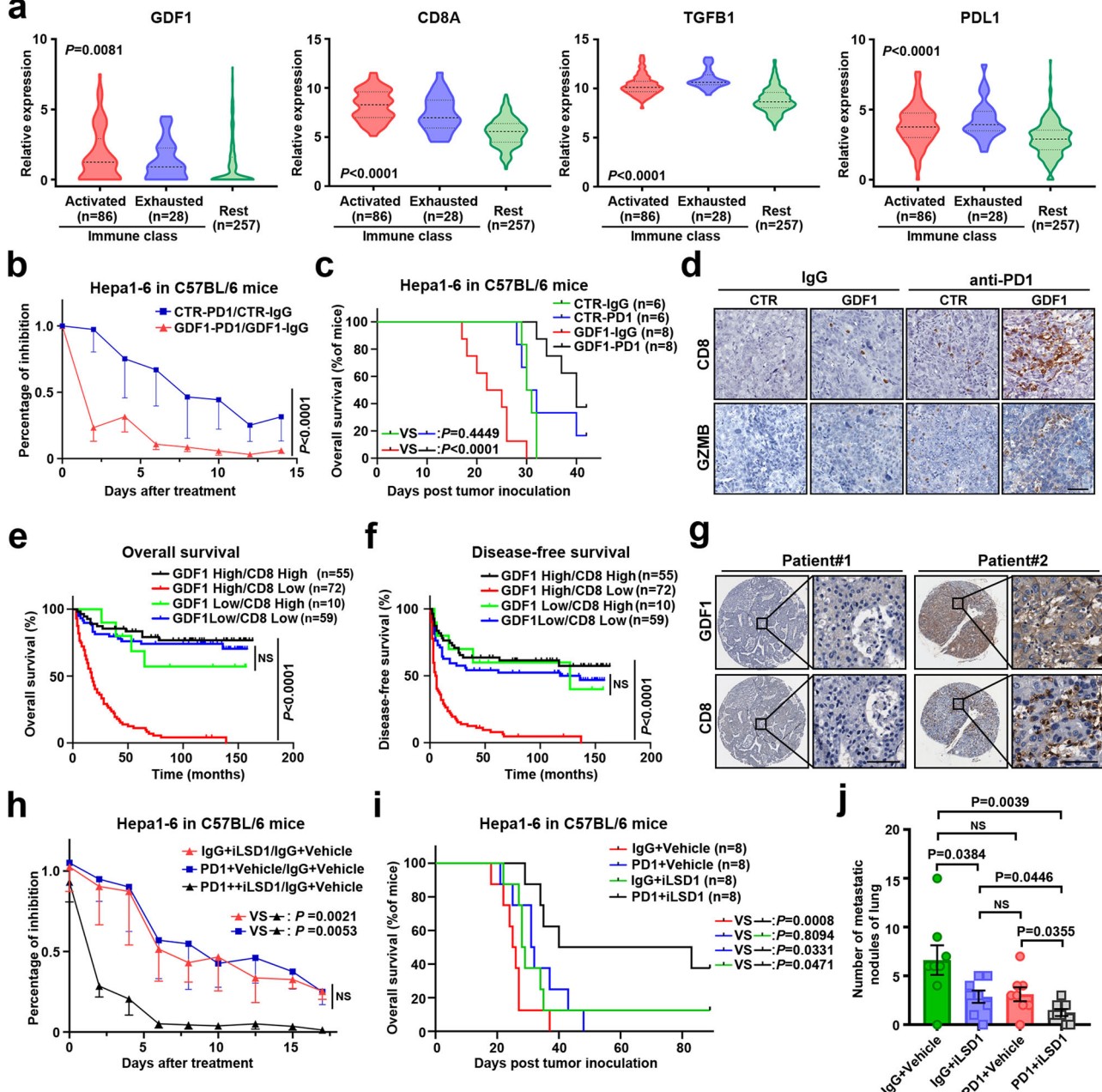

**Fig. 6 GDF1-induced tumour-lineage plasticity might sensitise HCC patients to anti-PD1 therapy. a** HCC patients from the TCGA–LIHC project were divided into three subgroups according to the specified classifiers. The relative expression of GDF1, CD8A, TGFB1, and PDL1 was shown in different subgroups of patients (one-way ANOVA test). **b** C57BL/6 mice bearing Hepa1–6 tumours transfected with mGDF1 or control vector were treated with anti-PD1 antibodies or control IgG at 100 μg/mouse every three days. The tumour-inhibition rates were detected by an in vivo bioluminescence imaging system (two-way ANOVA test, data are presented as mean values ± SEM, CTR-IgG: $n = 6$, CTR-PD1: $n = 6$, GDF1-IgG: $n = 8$, GDF1-PD1: $n = 8$). **c** Kaplan–Meier analysis of overall survival in C57BL/6 mice bearing Hepa1–6 tumours transfected with mGDF1 or control vector and treated with anti-PD1 antibodies or control IgG (log-rank test). **d** IHC staining of CD8 and GZMB in mouse orthotopic Hepa1–6 tumours transfected with mGDF1 or control vector and treated with anti-PD1 antibodies or control IgG. Scale bars represent 50 μm, $n = 3$ independent experiments. **e, f** IHC staining of GDF1 and CD8 + infiltrating T lymphocytes was performed in consecutive TMA slides. Kaplan–Meier analysis of overall survival (**e**) and disease-free survival (**f**) in HCC patients divided into four subgroups according to GDF1 and CD8 staining (NS, not significant, log-rank test). **g** Representative IHC staining of GDF1 and CD8 in consecutive TMA slides with 196 liver tumour tissues. Scale bars represent 50 μm. IgG, immunoglobulin G. PD1, anti-PD1 antibody. **h** C57BL/6 mice bearing Hepa1–6-CTR or Hepa1–6-GDF1$^{KO}$ cells were treated with LSD1 inhibitor, anti-PD1 antibody, or combination of both. For treatment with antibody, tumour-bearing mice were treated with anti-PD1 antibodies or control IgG at 100 μg/mouse every three days. For LSD1 inhibitor treatment, tumour-bearing mice were treated with 10 mg/kg of GSK-LSD1 or vehicle (4% DMSO in saline) each week (4 consecutive days followed by a 3-day holiday). The tumour inhibition rates were detected by an in vivo bioluminescence imaging system (NS, not significant, two-way ANOVA test, data are presented as mean values ± SEM, $n = 8$ mice per group). **i** Kaplan–Meier analysis of overall survival in C57BL/6 mice bearing Hepa1–6 tumours with different treatment groups (log-rank test). **j** The number of metastatic lung nodules in mice bearing Hepa1–6 tumours with different treatment groups. (NS, not significant, two-tailed independent Student's $t$-test, data are presented as mean values ± SEM, $n = 8$ mice per group). Source data are provided as a Source Data file.

observed in HCC and are closely associated with hepatic cancer stem cells[32]. In addition, the TGF-β superfamily, which comprises many other family members, including Activins, Nodal, bone morphogenetic proteins (BMPs), and GDFs, also plays crucial roles in both tumour development and dissemination[20]. In particular, the Nodal/Activin signalling-pathway components were found to be overexpressed in pancreatic cancer stem cells and to potentially increase the plasticity of tumour cells through ALK4/7 receptors[33]. GDF1 was first identified to regulate left/right patterning during development[34]. Later, it was found to interact with Nodal to form heterodimers and greatly potentiate Nodal signalling during stem cell differentiation[35]. However, there have been limited studies on the role of GDF1 in tumour plasticity. In the present study, we found that GDF1 is highly expressed in poorly differentiated HCCs and closely associated with poor patient prognosis. Functionally, we found that GDF1 suppressed HCC cell proliferation, which was in accordance with previous studies in gastric cancer[36]. However, overexpression of GDF1 strongly induced tumour dissemination and metastasis both in vitro and in vivo. This indicated a dual role of GDF1 in HCC malignant transformation, which is a typical characteristic of TGF-β signalling. Furthermore, we found that GDF1 induced the expression of liver progenitor markers but suppressed mature hepatic markers, indicating the enhanced cellular plasticity and retrodifferentiation of HCC. Interestingly, GDF1 also reactivated a broad panel of CTAs in HCC, which implied a potential opportunity for immune-based therapies. Of note, Inhibin alpha (INHA), encoding activin A, which ranked the second hit after GDF1 in the differential expressed genes of poorly differentiated HCCs, signals via ALK4 and SMAD2/3. We found that knockdown of ALK4 strongly potentiated the sphere formation in the absence or presence of GDF1. It is possible that activin A has a role in maintaining HCC tumour-lineage plasticity and this will be the subject of future studies.

The emergence of immune-checkpoint inhibitors is an evolutionary milestone for cancer therapy. However, the major challenge is to find the "ideal" patient based on biomarker profiling to maximise the therapeutic response. In this study, we found that GDF1 might be an indicator for effective immunotherapy. Although overexpression of GDF1 itself strongly induced tumour dedifferentiation and metastasis, a shift in cancer cellular lineages exposed an Achilles heel for immune surveillance, the CTAs. This was verified by an in vivo mouse model receiving anti-PD1 therapy and the clinical observation that infiltration of CD8+ T cells dramatically prolonged survival rates in GDF1-high HCC patients but not in GDF1-low subgroups. Interestingly, the ability to activate CTAs seems to be GDF1-specific. Conversely, treatment with TGF-β1 under the same conditions suppressed CTAs expression. Although SMAD2/3 can be phosphorylated by both GDF1 and TGF-β1 treatment, the absolute converse downstream effects indicated that SMAD2/3 may recruit different cofactors to regulate its target genes depending on different environmental situations. It is also possible that differential non-SMAD signalling pathways that are activated by GDF1 and TGF-β1 might contribute to the opposite effects on LSD1 expression. We noticed that recent in silico analyses indicated negative correlations between cancer stemness and pan-cancer immunity based on their established stemness indices[17]. Tumours are highly heterogeneous and comprise a hierarchy of cancer stem cells at different developmental stages with distinctive characteristics[8,37]. We believe that additional precise stratification of cancer stemness and differentiation status might help more accurately predict the immune therapeutic response in specified cancer types. TGF-β signalling is a critical driver of immune evasion mainly through remodulation of immune cells in the tumour microenvironment[38]. In contrast to TGF-β, GDF1 was found to

be associated with an active immune response in this study. This was evidenced by the elevated expression of GDF1 in the immune active subgroup and the significant correlation with CD8+ infiltrating cytotoxic T cells. Unlike TGF-β, which is universally expressed in both normal and tumour tissues, the expression of GDF1 was restricted to high-grade HCCs and associated with the reactivation of CTAs. In addition, GDF1 binds to different receptors than TGF-β, which might also lead to completely different biological properties. The final antitumour immunity response depends on joint forces from multiple factors, including tumour cells, immune cells, and crosstalk in the microenvironment. Further investigation of the effects of GDF1 on immune cells in the tumour microenvironment might better reveal the roles of GDF1 in antitumour immunity. Considering that GDF1 can activate a broad panel of CTAs, the most potential mechanism is through epigenetic regulation. From our screening data, we found GDF1 can actually influence series of epigenetic regulators. Although we does not preclude the possibility that GDF1 can potentially affect other epigenetic regulators, LSD1 is the most significantly affected in the current study. Epigenetic therapy has recently shown promising synergistic effects with immune therapy[21]. LSD1 ablation has already been found to stimulate antitumour immunity and enable checkpoint blockade in a melanoma mouse model[39]. Our current study further supported the use of LSD1 inhibitors in combination with immune check point inhibitors (ICI) further in the clinic. While GDF1 might potentially serve as an indicator for ICI therapy, considering its strong association with tumour immunogenicity. Although LSD1 inhibitor could enhance CTAs expression in GDF1-negative cells, we have evidences that the activation of CTAs is much stronger in the presence of GDF1. The actual mechanism is still not clear, however, it is possible that the presence of GDF1 might prime the tumour microenvironment to favour LSD1 in regulating tumour immunogenicity. We tried to test the therapeutic effects in a spontaneous liver tumour mice model, e.g., DEN-induced hepatocarcinogenesis. However, we found that the DEN-induced liver tumour does not have GDF1 activation. This indicated that activation of GDF1 might be restricted to certain subtypes of liver tumour. Instead, we established a CRISPR-Cas9-mediated GDF1-knockout Hepa1–6 xenograft mice model. The poor tumorigenic ability of GDF1^KO Hepa1–6 cells in C57 mice further confirmed that GDF1 is critical for HCC tumorigenesis and metastasis. In the control GDF1^WT group, combination of LSD1 inhibitor with anti-PD1 antibody showed significant superior therapeutic advantages compared with single treatment groups in both suppressing tumour growth and metastasis, and prolonging overall survival. Taken together, inhibition of LSD1 might boost antitumour immunity and widen the therapeutic window for immune checkpoint inhibitors in HCC patients from a clinical translational perspective.

## Methods

**Mice, tumour specimens, and cell lines.** Mice were housed in a pathogen-free laboratory animal unit (LAU) at Guangzhou Medical University. All animal experiments were approved by the review board of Guangzhou Medical University. Studies using human tissues were reviewed and approved by the Committees for Ethical Review of Research involving Human Subjects of Sun Yat-Sen University and The University of Hong Kong. All patients gave written informed consent for the use of their clinical specimens for medical research. Detailed clinical–pathological variables of the HKU cohort, TCGA cohort, and the tissue microarray according to the REMARK reporting guidelines and the EASL Guidelines are listed in Supplementary Table 1 and Supplementary Table 3. Kaplan–Meier curves and log-rank test were used in overall survival and disease-free survival studies. Cox regression and logistic analyses were used to assess the independent prognostic factors using SPSS v25 (IBM, Inc., Chicago, IL, USA). A P value of less than 0.05 was considered statistically significant. The HCC cell lines PLC-8024, Huh7, Hep3B, and HepG2 were purchased from ATCC, and HCC cell lines CLC1, CLC2, CLC5, CLC11, CLC13, and CLC16 were kind gifts by Prof. Li-

Jian Hui from the Shanghai Institute of Life Science, Chinese Academy of Sciences. All the cells used were authenticated and tested for mycoplasma.

**Cell culture.** All HCC cells were cultured in high-glucose Dulbecco's modified Eagle's medium (DMEM) (Gibco BRL, USA) supplemented with 10% foetal bovine serum (FBS) (Gibco BRL, NY). The 293FT cells used for viral packaging were cultured in high-glucose DMEM with 10% FBS, 6 mM L-glutamine, 1 mM sodium pyruvate (Invitrogen, USA), and 0.1 mM nonessential amino acids (NEAAs) (Invitrogen, USA). To maintain the expression of SV40 large T antigen in 293FT cells, geneticin at a dose of 500 µg/mL was also added (Roche, Germany). All the cells were kept at 37 °C in a humidified incubator containing 5% carbon dioxide.

**RNA extraction and quantitative real-time PCR.** Total RNA was extracted using TRIzol Reagent (Life Technologies, CA), and reverse transcription was performed using an Advantage RT-for-PCR Kit (Takara, JPN) according to the manufacturer's instructions. For qPCR analysis, aliquots of double-stranded cDNA were amplified using a SYBR Green PCR Kit (Vazyme, CHN) and an ABI PRISM 7900 Sequence Detector. For cell lines, the relative gene expression is given as $2^{-\Delta CT}$ ($\Delta CT = CT(gene) - CT(18S)$) and normalised to the relative expression that was detected in the corresponding control cells. For clinical samples, we calculated the relative expression of target genes in clinical HCCs and their matched nontumour specimens by the formula $2^{-\Delta CT}$ ($\Delta CT = CT(target\ genes) - CT(18S)$) and normalised them to the average relative expression in all of the nontumour tissues, which was defined as 1.0. An RT2 Profiler PCR Array (QIAGEN, Germany) was used to profile the gene expression of selected chemokines and cytokines. The detailed probe information was listed in Supplementary Table 4.

**Lentivirus-mediated gene overexpression or knock down.** For GDF1 overexpression assay, human GDF1 and mouse GDF1 plasmids (Cyagen Biosciences, China) were transfected into the 293FT cell line. Virus-containing supernatants were collected for subsequent transduction into PLC-8024 and Hepa1–6 cells. In all, 1 µg/mL of puromycin (Invitrogen, CA) was used to select stably transduced cells. For the knock down assay, the viruses containing shRNAs targeting ALK4, ALK5, ALK7, SMAD2, and SMAD3 (Gene Chem, China) were packaged and transduced into HCC cell lines according to the manufacturer's instructions. Western blot was performed to confirm the knockdown efficiency. Stable human GDF1 and mouse GDF1 knockout cells were purchased from Cyagen Biosciences. Gene-knockout efficiency was validated by Western blot and DNA Sanger sequencing. The shRNA and gRNA target sequences used in the study were listed in Supplementary Table 5.

**RNA sequencing, databases, and statistical analysis.** RNA sequencing was performed at Shanghai Shenggong Biotech Co., Ltd. For input material of RNA sample preparations, a total amount of 2 µg of RNA per sample was used. VAHTSTM mRNA-seq V2 Library Prep Kit for Illumina® was used to generate sequencing libraries. Index codes were added to attribute sequences to each sample. Briefly, poly-T oligo-attached magnetic beads were used to isolate mRNA from total RNA. RNA fragmentation was performed using divalent cations under elevated temperature in VAHTSTM First Strand Synthesis Reaction Buffer (5X). M-MuLV Reverse Transcriptase (RNase H-) was used to synthesise first-strand cDNA with random hexamer primer. DNA polymerase I and RNase H was used to synthesise second-strand cDNA. The remaining overhangs were converted into blunt ends with exonuclease/polymerase activities. Adaptor was ligated to prepare for library after adenylation of 3′ ends of DNA fragments. The library fragments were purified with AMPure XP system (Beckman Coulter, USA) to select cDNA fragments in the range of 150–200 bp. Size-selected, adaptor-ligated cDNA was mixed with 3 µl of USER Enzyme (NEB, USA) at 37 °C for 15 min followed by 5 min at 95 °C before PCR. Then PCR was performed with Phusion High-Fidelity DNA polymerase, using universal PCR primers and Index (X) primers. PCR products were purified (AMPure XP system) and library quality was checked on the Agilent Bioanalyzer 2100 system. Paired-end sequencing of the library was performed on the HiSeq XTen sequencers (Illumina, CA). FastQC (version 0.11.2) was used for evaluating the quality of sequenced data. Raw reads were filtered by Trimmomatic (version 0.36), then the remaining clean data were used for further analysis. These high-quality reads were aligned against the human genome assembly (National Center for Biotechnology Information build 37.1/hg19) using HISAT2 (version 2.1.0) with the RefSeq refGene annotation, which was downloaded from the University of California, Santa Cruz (UCSC) Genome Browser. RSeQC (version 2.6.1) was used for statistics of the alignment results and analysis of the redundancy sequences and the distribution of insert fragments. Qualimap (version 2.2.1) was used to check the homogeneity distribution and analysis of the genome structure. BEDTools (version 2.26.0) was used for statistical analysis of the gene-coverage ratio and the distribution of sequenced sequence on chromosome. Transcript abundances were quantified by StringTie (version 1.3.3b), which was used to calculate transcripts per million (TPM) of both protein-coding genes and lncRNAs in each sample. For the samples without biological repetition, TMM was used to standardise the read-count data, and then DEGseq (version 1.26.0) was used for differential gene expression analysis. In order to obtain the significant

differential genes, the screening conditions were set as follows: q-value < 0.05 and difference-multiple log2FoldChange > 1. 1811 differentially expressed genes were identified, including 545 upregulated genes and 1266 downregulated genes. Detailed QC metrics were listed in Supplementary Fig. 9. The expression summary file was shown in Supplementary Data 2.

**Gene set enrichment analysis.** Gene Set Enrichment Analysis was performed using GSEA software (http://www.broadinstitute.org/gsea/) with permutation type: "gene set", metric for ranking genes: "log2_Ratio of Classes", enrichment statistic: "weighted", and numbers of permutations: "1000"[40]. The CTA gene sets, including GAGE family members and other CTAs, were obtained from the publicly-maintained and recognised dataset CTDatabase (http://www.cta.lncc.br/modelo.php). A total of 276 CTA genes were used in the analysis. The analysis was not performed with a preranked gene list. Genes were listed in Supplementary Data 3.

**In vitro functional assays for tumorigenicity.** For the cell-proliferation assay, cells were seeded in 96-well plates at a density of 500 cells/well. The cell growth rate was detected using a Celigo Imaging Cytometer (Nexcelom, MA). For the foci-formation assay, cells were seeded in 6-well plates at a density of 500 cells per well. For the sphere-formation assay, cells were seeded in 24-well ultralow-attachment plates at a density of 500 per well in a suspension of 300 µl of serum-free DMEM/F12 medium supplemented with 20 ng/mL human recombinant EGF, 10 ng/mL human recombinant bFGF, 4 µg/mL insulin, B27, 500 units/ml penicillin, and 500 µg/mL streptomycin in poly (2-hydroxyethyl methacrylate)-coated 24-well plates. Cells were replenished with 30 µl of supplemented medium every second day. The clones formed were counted in 2 to 3 weeks. For the Transwell migration assay, Transwell membranes (diameter: 24 mm, pore size: 0.4 µm, Corning Costar 3450) were used. For the Transwell migration assay, Transwell membranes (diameter: 6.5 mm, pore size: 8 µm, Corning Costar 3422) were used. Cells were seeded in the top chambers at a density of $1.0 × 10^5$ cells/well in FBS-free medium, while medium containing 10% FBS was applied to the lower chamber. After 24 h, the membrane was fixed in situ with 4% paraformaldehyde, and cells were stained with crystal violet for light microscopy. For the Transwell invasion assay, similar membranes coated with Matrigel (diameter: 6.5 mm, pore size: 8 µm, Corning Costar 354480) were used. Cells were seeded in the top chambers at a density of $2.0 × 10^5$ cells/well in FBS-free medium for 42 h to determine invasive potential.

**In vitro 3D organoid culture of primary liver tissue.** Primary liver tissues were obtained from six-week-old C57BL/6 mice. Cells were isolated and cultured in an organoid culture system according to published protocol[41]. Briefly, the tissue was minced and incubated at 37 °C with the digestion buffer. Digestion was stopped after there were no visible pieces of tissue remaining and filtered through a 100 µm nylon-cell strainer (Falcon, USA). The cells were resuspended in material (R&D Systems, USA) and seeded in a 24-well plate. The isolation medium with normal liver-expansion medium was replaced after 3–4 days of culture. Expansion medium was changed twice a week, and cultures were split upon the attainment of dense culture. The organoids were confirmed at the histological level. The organoids of normal liver tissues presented cyst-like hollow structure. Lentivirus-mediated transfection of mGDF1 or control vector was performed in organoid cultures and immunofluorescent staining of Ki67 was performed according to published protocol[41].

**In vivo tumorigenic and metastatic assay.** For the subcutaneous tumorigenic assay, PLC-8024-CTR or PLC-8024-GDF1 ($2 × 10^6$ cells/mouse) was injected into the dorsal flank of six-week-old BALB/c nude mice subcutaneously. The tumour volumes were assessed three times per week and were calculated according to the formula $V = 0.5 × L × W^2$. When the tumour burden was >1500 mm3 or the animal's weight reduced >20%, mice were euthanized, and the tumours were collected to generate paraffin-embedded sections, which were used for further histological and immunohistochemical analysis. For the intrasplenic injection model, PLC-8024-CTR or PLC-8024-GDF1 ($2 × 10^6$ cells/mouse) was injected into the spleen of the tested nude mouse (six-week-old). After 6 weeks, the mice were euthanized, and the mouse livers and lungs were collected to generate paraffin-embedded sections that were used for further histological and immunohistochemical analysis.

**Immunohistochemical staining (IHC), immunofluorescent staining (IF), western blot (WB), antibodies, and inhibitor.** Paraffin-embedded tissue sections were deparaffinized and rehydrated according to the manufacturer's instructions. Slides were immersed in 10 mM citrate buffer and boiled for 15 min for antigen retrieval. After rinsing in PBS, the tissue slides were incubated with primary antibody at 4 °C overnight in a moist chamber. Then biotinylated general secondary antibody was added and incubated for 1 h at room temperature. After rinsing in PBS for 5 min, streptavidin-peroxidase conjugate was added for incubation at room temperature for 15 min. Finally, a 3,5-diaminobenzidine (DAB) Substrate Kit (Dako, Carpinteria, CA) was used for colour development followed by Mayer's haematoxylin counterstaining. IF and WB were performed according to the manufacturer's standard protocols. IHC slides were scanned using an Aperio CS2 Digital Pathology Scanner (Leica, Germany) at ×20 magnification. Positive

staining was assessed using a 5-point scoring system: 0 (0 positive cells), 1 (<10 positive cells), 2 (10–35% positive cells), 3 (36–70% positive cells), and 4 (>70% positive cells). Meanwhile, the intensity of positive staining was also assessed: 0 (negative), 1 (weak), 2 (moderate), and 3 (strong). The expression of target-protein index was calculated as follows: expression index = (positive score) × (intensity score). Optimal cutoff values for this score system was identified as follows: high expression was defined as an expression index score of ≥4, and low expression was defined as an expression-index score of <4. The antibodies used in this study include: Recombinant Human GDF-1 Protein (R&D, 6937-GD), Recombinant Human TGF-β1 Protein (R&D, 240-B), GDF1 (IF, 1:100, R&D, MAB6937), GDF1 antibody (IHC,1:200, WB, 1:1000, Biorbyt, orb33665), GDF1 (WB, 1:500, IHC, 1:400, Aviva System Biology, OACD03794), GDF1 (IF, 1:100, Abbexa, abx103259), CD8 (IHC, 1/500, Abcam, ab93278), CD8 (IHC, 1/2000, Abcam, ab209775), Granzyme B (IHC, 1/100, Abcam, ab4059), Ki67 (IF, 1/40, R&D, AF7649), Cytokeratin 19 (IHC/IF, 1/500, WB, 1/1000, Abcam, ab52625), SOX9 (IHC/IF, 1/ 500, WB, 1/1000, Abcam, ab185966), AFP (IF, 1/100, IHC, 1:200, WB, 1/1000, Proteintech, 14550-1-AP), AFP (WB, 1/1000, Abcam, ab46799), EPCAM (IHC/IF, 1/100, WB, 1/1000, Abcam, ab92469), TTR (IF, 1/100, Abcam, ab75815), HNF-4-alpha (IHC, 1/2000, WB, 1/1000, Abcam, ab181604), GAGE12E (IHC, 1/100, WB, 1/1000, Aviva System Biology, OAAB07499), LSD1 (WB, 1/10000, Abcam, ab129195), GAPDH (WB, 1/2000, Cell Signalling Technology, 51332), SMAD2/3 (WB, 1/1000, CHIP:1:100, Cell Signalling Technology, 8685), Phospho-SMAD2/3 (WB, 1/1000, Cell Signalling Technology, 8828), SMAD2 (WB, 1/10000, Abcam, ab40855), SMAD3 (WB, 1/1000, Abcam, ab208182), ALK4 (WB, 1/10000, Abcam, ab109300), ALK5 (WB, 1/1000, R&D, AF3025), ALK7 (WB, 1/2000, Novus Bio-logicals, NBP1-50659), Anti-rabbit IgG, HRP-linked Antibody (Cell Signalling Technology, 7074, 1:5000), and Anti-mouse IgG, HRP-linked Antibody (Cell Signalling Technology, 7076, 1:5000). The LSD1 inhibitor used in this study was purchased from MCE (GSK-LSD1 dihydrochloride, HY-100546A).

**Luciferase-reporter assay**. Luciferase-reporter assay was performed using the Dual Luciferase Assay System (Promega, USA) according to the manufacturer's instructions. PLC-8024 cells treated with rGDF1 or rTGF-β1 or vehicle control were lysed with a lysis buffer. Relative promoter activity of LSD1 was measured using a Synergy Mx Multi-Mode Reader (Biotek, USA) and normalised according to the Renilla activity.

**Chromatin immunoprecipitation (ChIP) assay**. PLC-8024 cells were treated with rGDF1 at 50 ng/mL or rTGF-β1 at 10 ng/mL or vehicle control for 30 min. The cells were washed twice with cold PBS and cross-linked in 16% formaldehyde (w/v) at room temperature for 10 min. Then, the cells were resuspended in glycine solution (10×) to a final concentration (1×). After washing with cold PBS, cells were collected by centrifugation at 3000 g for 5 min. Cell lysis and MNase digestion were performed using the Agarose chip kit (Thermofisher, US) according to the manufacturer's standard protocols. Supernatants were incubated with SMAD2/3 antibodies (Cell Signalling Technology, 8685) or isotype control IgG (Cell Signalling Technology, 2729) overnight at 4℃. Protein A/G beads were used to capture antibody-DNA complexes, and the DNA was purified and detected by qPCR analysis. Primers used in chip-qPCR analysis were listed in Supplementary Table 4.

**Live animal imaging and in vivo therapeutic treatment of mice with anti-PD1 antibody**. To establish a luciferase-labelled orthotopic HCC model, Hepa1-6-CTR or Hepa1-6-GDF1 cells (1 × 10⁶ cells/mouse) stably expressing firefly luciferase were injected into the spleens of six-week-old C57BL/6 mice. Tumour formation and metastasis were evaluated with the IVIS Lumina XRMS Series III system (PerkinElmer, USA). Mice were intraperitoneally injected with 15 mg/mL D-luciferin (Gold Biotech, USA) at 150 mg/kg per mouse 5 min before imaging. Hepa1–6-CTR or Hepa1-6-GDF1 mice were randomly divided into two groups. Once the bioluminescence signal reached 10⁵ lux/sec, tumour-bearing mice were treated with an anti-PD1 antibody at a dosage of 100 µg/mouse (Bio X Cell, USA, BP0146) or a rat IgG2a isotype control (Bio X Cell, USA, BP0089) every 3 days. The bioluminescence signals were monitored every 2 days. When the biolumi-nescence signal reached 3 × 10⁷ lux/sec or the animal's weight reduced >20%, mice were euthanized. To establish a luciferase-labelled metastatic HCC model for in vivo combination treatment of mice with LSD1 inhibitor and anti-PD1 antibody, Hepa1-6-CTR or Hepa1–6-GDF1^KO cells (1.5 × 10⁶ cells/mouse) stably expressing firefly luciferase were injected into the spleens of six-week-old C57BL/6 mice. Tumour formation and metastasis were evaluated with the IVIS Lumina XRMS Series III system. Once the bioluminescence signal reached 10⁵ lux/sec, mice were randomised into 4 groups. For treatment with antibody, tumour-bearing mice were treated with an anti-PD1 antibody at a dosage of 100 µg/mouse or a rat IgG2a isotype control every 3 days. For LSD1 inhibitor treatment, mice were treated with 10 mg/kg of GSK-LSD1 (MCE, HY-100546A) or vehicle (4% DMSO in saline) each week (4 consecutive days followed by a 3-day holiday, i.p.). The bioluminescence signals were monitored every 2 days. When the bioluminescence signal reached 3 × 10⁷ lux/sec or the animal's weight reduced >20%, mice were euthanized. To establish a GDF1^KO-metastatic HCC model, Hepa1-6-GDF1^KO cells (3.5 × 10⁶ cells/mouse) stably expressing firefly luciferase were injected into the spleens of six-

week-old C57BL/6 mice. Tumour formation and metastasis were monitored with the IVIS Lumina XRMS Series III system every two days.

**Statistical analysis**. Statistical analysis was performed using GraphPad Prism 9.0. Cox regression analysis and χ² test were used to assess the association with overall survival using SPSS v25 (IBM Inc.). All statistical tests are two-tailed with statistical significance defined as $P < 0.05$. Independent Student's t-test was used to compare the mean expression level of two different groups, and one-way analysis of variance (ANOVA) test was used to compare means between three and more subgroups. All survival curves were plotted using Kaplan–Meier survival analyses and log-rank test. Pearson's χ² test was used for the analysis of clinical–pathological features and correlation of gene expression. Figures in this work were generated using Adobe Photoshop CC2019.

**Reporting summary**. Further information on research design is available in the Nature Research Reporting Summary linked to this article.

## Data availability
The RNA-seq data generated in this study have been deposited in the GEO repository under accession code GSE188582. Publicly available databases and reference genome assembly used were as follows: ENCODE [https://www.encodeproject.org], Human genome assembly GRCh37 (hg19) [https://ftp.ncbi.nlm.nih.gov/genomes/all/annotation_releases/9606/105.20201022/GCF_000001405.25_GRCh37.p13/]. The data in Supplementary Fig. 2e and Supplementary Fig. 4f used in this study are available from the Linkedomics web server database [http://www.linkedomics.org/login.php]. The remaining data are available within the article, Supplementary Information, or Source Data file. Source data are provided with this paper.

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

## Acknowledgements

The HCC cell lines CLC1, CLC2, CLC5, CLC11, CLC13, and CLC16 were kind gifts by Prof. Li-Jian Hui from the Shanghai Institute of Life Science, Chinese Academy of Sciences. This work was supported by the National Natural Science Foundation of China (81702400), Guangzhou Key Medical Discipline Construction Project Fund, Guangdong Basic and Applied Basic Research Foundation (2019A1515011787), Guangdong Province Pear River Talent Program (2017), and Guangdong Province Universities and Colleges Pear River Scholar Funded Scheme (2018).

## Author contributions

M. Liu and W. Cheng initiated and designed the experiments; W. Cheng performed the experiments with input from H. L. Li, X. F. Zhang, L. Xing, Y. X. Mo, M. M. Li, H. Q. Cui, X. G. Chen, and Z. M. Cao; W. Cheng analysed and interpreted the data with assistance from F. E. Kong, and W. J. Zhu and Y. Zhu performed bioinformatics analysis; S. Y. Xi, Y. F. Gong and Y. Q. Tang provided the clinical samples and patient information; Y. Zhang, N. F. Ma and X. Y. Guan provided valuable comments; M. Liu and W. Cheng wrote the paper; and all authors reviewed and approved the paper.

## Competing interests

The authors declare no competing interests.

## Additional information

**Peer-review information** *Nature Communications* thanks Peter ten Dijke, Hyam Leffert and the other, anonymous, reviewer(s) for their contribution to the peer review of this work. Peer-reviewer reports are available.

