## [Peer Review File · Nature Communications]

Reviewers' Comments:

Reviewer #1:

Remarks to the Author:

In this paper the authors identify GDF1 was a factor that correlates with poor progression of hepatocellular carcinoma. While GDF1 inhibits HCC proliferation it promotes metastasis and lineage plasticity. GDF1 stimulates the expression of cancer testis antigens, which promotes the immunogenicity of HCC cells to immune therapies. The ALK4/7/Smad3 pathway is claimed to repress the expression of epigenetic regulator LSD1 to stimulate the expression of CTAs.

The reported findings are novel and (clinically) interesting.

Comments:

The statement that GDF1 functions via ALK4/7 and SMAD3 is not supported by the data. LY2157299 is not a selective compound that only targets ALK5. It also inhibits ALK4 (and likely ALK7). See for example the following paper:
<https://www.ncbi.nlm.nih.gov/pmc/articles/PMC5805504/>

SIS2 is not a selective inhibitor of SMAD3. Its has many off targets. Also the claim that it inhibits SMAD3 is very weak.

I strongly recommend that the authors use siRNA (or shRNA) to target specific ALK4, ALK5 or ALK7, SMAD3 and SMAD2 to demonstrate involvement of ALK4, ALK7 and SMAD3 (and not ALK5 and SMAD2).

2. GDF1 strongly stimulates CTAs expression and inhibits LSD1 expression. What is known about the expression in HCC clinical samples? The authors could perform data mining of publicly available data bases.

3. Page 12. I do not understand how the Figure F4D demonstrates "We found the expression of almost all the tested CTAs increased progressively from patients with progressive disease to subgroups with partial response and complete response". The authors should better clarify this figure.

4. End of results statement (and also end of discussion makes a similar statement): Inhibition of LSD1 in GDF1 positive HCC tumors can strongly boost the expression of CTAs. Yes, that may be true (but authors do not show this with actual data), but as GDF1 already decreases LSD1 expression, would it be not more interesting to see if GDF1 negative tumors can be come more immunogenic by inhibiting LSD1?

5. It is unclear how GDF1 represses LSD1 expression? Can SMAD3 directly be recruited to LSD1 promotor and repress transcription? This reviewer is also intrigued regarding the difference between GDF1 and TGF-beta. Does TGF-beta (that also signals via SMAD3) not repress LSD1 expression?>

6. Include molecular weight markers on all the Western blots.

Reviewer #2:

Remarks to the Author:

In their manuscript, Wei Cheng et al. suggest that GDF1-induced tumor plasticity drives cancer-testis antigen presentation, with implications for GDF1 expression as a biomarker for extending the therapeutic window for immunotherapy. I was asked to provide my expert opinion on the technical aspects of RNA sequencing in this study, and notified that all other areas are covered by alternate reviewers. I am therefore limiting my response solely to this facet of the study.

RNA sequencing was performed on PLC-8024 cells transfected with GDF1 or control vectors. This was used to explore differentially expressed genes, illustrated in Figure 5A and S3A and briefly described in-text. I reviewed the full description of the technical aspects of the RNA sequencing

experiment from this work, and will reproduce it here:

"RNA sequencing was performed in Shanghai Shenggong biotech co. ltd."

This information is insufficient for me to gauge the rigor and suitability of the experiment in supporting the results described in this study. There is no data availability statement, and no data (including summarized expression values) from the RNA sequencing experiment described in this work (notably there is the Table S1 dataset provided for the related rtPCR experiments). For me to properly evaluate the technical merit of this aspect of the work, the additional information should be provided:

- Library preparation description and sequencing technology
- QC metrics, including at a minimum RIN values, alignment quality distribution, percentage uniquely mapped reads, 5'/3' bias distribution, and insert size distribution (if applicable).
- Read alignment and expression estimation software used, and associated parameters
- Expression summary files

It is also worth noting that the above description of the RNA sequencing experiments preceded the following:

"DAVID bioinformatics resources (<https://david.ncicrf.gov/home.jsp>) was used for gene ontology analysis."

This is likely the foundation for the following statement from the results:

"the most differentially expressed genes were enriched in the cancer testis antigens, which include the GAGE family members"

However, without further description of the analyses undertaken to generate this result, the subset of GAGE family gene symbols overlaid on panel 5A and S3A appear cherry-picked to support the narrative of this work. Many members of the GAGE family—and other CTAs—are not mentioned or described.

A formal gene set enrichment analysis of all CTAs or GAGE-family CTAs (with gene family members as defined by publicly-maintained and recognized datasets, such as those at <http://www.cta.lncc.br/modelo.php>) might go a long way towards enforcing this stated result from the RNA sequencing data.

Reviewer #3:

Remarks to the Author:

This manuscript is potentially of great importance. For it suggests new ways to identify human liver cancer patients (from the serum biomarker, GDF1) amenable to specific molecular immunotherapy treatments that might augment survival significantly.

The major claims of this paper are novel and, without question, will be interesting to others in the hepatology community as well as the wider fields of human cancer and immunotherapy.

However, the work and its conclusions, for the most part convincing (the major primary observations are made with tissue biopsies from human patients suffering from hepatocellular carcinoma [HCC]), is based upon evidence of association.

It is suggested therefore that the work would be strengthened by inclusion of:

1. Immunohistochemical staining of fetal mouse livers (Fig. 3A) and by improved images of GDF1 immunohistochemical staining at negative, low, moderate and high staining intensities in HCC (Fig. S1, C; the latter images appear to have been made from tissue sections subjected to fixation artifact).
2. Partial hepatectomy and primary hepatocyte culture studies with wild type mice (or rats). Retro-differentiation phenomena occur in both models, so GDF1 expression is expected. The herculean work effort reported here relied upon HCC cell lines, not primary HCC cells. For example, will GDF1 block proliferation of normal hepatocytes?
3. DEN-chemical hepatocarcinogenesis and immunotherapy studies with wild type (control) and hepatocyte-specific GDF1 knockout mice would greatly strengthen and prove the paper's major conclusions without doubt.

The paper will definitely influence thinking in the field. For example, is it by chance that GDF1 – a protein involved in the determination of left-right symmetry during development – functions via hepatic ALK4/7-SMAD3 signaling cascades? What are the additional mechanisms by which GDF1 reactivates cancer testis antigens in HCCs?

Statistical analyses are appropriate and valid throughout.
Given the level of detail provided, the experiments should be reproducible.

The manuscript should be carefully rewritten as it is filled throughout with English language errors.

Reviewed by HL Leffert.

Point-by-point response:

Reviewer #1 (Remarks to the Author):

In this paper the authors identify GDF1 was a factor that correlates with poor progression of hepatocellular carcinoma. While GDF1 inhibits HCC proliferation it promotes metastasis and lineage plasticity. GDF1 stimulates the expression of cancer testis antigens, which promotes the immunogenicity of HCC cells to immune therapies. The ALK4/7/Smad3 pathway is claimed to repress the expression of epigenetic regulator LSD1 to stimulate the expression of CTAs.

The reported findings are novel and (clinically) interesting.

Comments:

1. The statement that GDF1 functions via ALK4/7 and SMAD3 is not supported by the data. LY2157299 is not a selective compound that only targets ALK5. It also inhibits ALK4 (and likely ALK7). See for example the following paper: <https://www.ncbi.nlm.nih.gov/pmc/articles/PMC5805504/>

SIS2 is not a selective inhibitor of SMAD3. It has many off targets. Also, the claim that it inhibits SMAD3 is very weak.

I strongly recommend that the authors use siRNA (or shRNA) to target specific ALK4, ALK5 or ALK7, SMAD3 and SMAD2 to demonstrate involvement of ALK4, ALK7 and SMAD3 (and not ALK5 and SMAD2).

Our reply:

Thanks for the reviewer's suggestions, we have designed shRNAs specifically targeting different critical components of the GDF1 signaling pathway for mechanism studies. As shown in the results, the enhanced tumor cell invasion and self-renewal ability induced by GDF1 was not affected by knocking down of ALK4 or ALK5. Instead, knock down of ALK7 significantly abolished cell invasion and self-renewal induced by GDF1 overexpression (Fig. 4, A, B, and C). These findings indicated that GDF1 mainly functions through the cell receptor ALK7 to activate downstream targets and the malignant phenotypes. To investigate the involvement of SMAD2 and SMAD3, shRNAs specifically targeting these two genes were designed and stably transfected into HCC cells. We found knock down of either SAMD2 or SMAD3 can abolish the increased liver progenitor biomarkers and decreased mature hepatic markers induced by GDF1 overexpression (Fig. 4, E and F). In addition, knock down of either SAMD2 or SMAD3 also abolished the decreased LSD1 and increased GAGE12E expression induced by GDF1 (Fig. S5, H, I). Considering that SAMD2 and SAMD3 usually forms complexes in regulating gene transcription, it is possible that both SAMD2 and SAMD3 are required for tumor lineage plasticity and downstream signaling induced by GDF1. This part of data has been incorporated into the "Results" section (1st paragraph of page 9), Fig. 4, A, B, C, E, F, as well as in Fig. S3, A, B, Fig. S5, H, I.

2. GDF1 strongly stimulates CTAs expression and inhibits LSD1 expression. What is known about the expression in HCC clinical samples? The authors could perform data mining of publicly available data bases.

Our reply:

In accordance with reviewer's suggestions, we have checked the expression of GDF1 and series of CTAs in the LIHC project of the TCGA database. As shown in Fig. S4F, although the expression of GDF1 showed the trend of positive correlation with representative GAGEs, it didn't reach statistical significance. This might be due to the zero expression of GAGEs in a certain proportion of HCC patients. To further test whether GDF1 affects other CTAs expression in HCC. The expressions of representative MAGE and LY6 family members were examined by qPCR in HCC cells. The results showed that overexpression of GDF1 can also significantly increase the expression of MAGE and LY6 family members (Fig. S4E). In addition, significant positive correlation between GDF1 and representative MAGE and LY6 family members was also found in the TCGA database (Fig. S4F). These findings indicated that GDF1 can activate a broad panel of CTAs in HCC. The data have been incorporated into the "Results" section, (2nd paragraph of page 10), as well as in Fig. S4, E, F.

3. Page 12. I do not understand how the Figure F4D demonstrates "We found the expression of almost all the tested CTAs increased progressively from patients with progressive disease to subgroups with partial response and complete response". The authors should better clarify this figure.

Our reply:

In this figure, data was derived from a publicly available database with melanoma patients treated with anti-PD1 therapy. The patients were divided into 3 subgroups according to their treatment responses (Group1: progressive disease, blue column; Group2: partial response, green column; Group3: complete response, red column). The columns in the figure with different colors represents the average gene expression value of the indicated CTAs in the 3 subgroups. Due to limited spaces, different columns representing the expression of the same gene (e.g. GAGE1) were overlapped. The average expression value of GAGE1 increased progressively from the patients in Group1 to that in Group 2 and Group3. This indicated that patients with complete response to anti-PD1 therapy have higher expression of GAGE1 than patients with partial response or progressive disease. Similar trend could be seen in other CTAs. Actually, this is only an indirect evidence to support the close association between CTAs expression and clinical response in melanoma patients. Since anti-PD1 antibody was just recently approved by FDA to treat sorafenib refractory HCC patients, no clinical data is available regarding responses to anti-PD1 therapy currently either in our cohort or in publicly available databases. Considering that the data from melanoma might not be able to rigorously reflect the actual situation in HCC, we

decided to remove this part in the revised manuscript to prevent misleading the authors. The discussion and literature review of close association between CTAs and anti-PD1 therapeutic response were retained.

4. End of results statement (and also end of discussion makes a similar statement): Inhibition of LSD1 in GDF1 positive HCC tumors can strongly boost the expression of CTAs. Yes, that may be true (but authors do not show this with actual data), but as GDF1 already decreases LSD1 expression, would it be not more interesting to see if GDF1 negative tumors can become more immunogenic by inhibiting LSD1?

Our reply:

Thanks for the reviewer’s suggestions. This is actually an important question regarding the dependency of GDF1 in LSD1-regulated tumor immunogenicity. We treated PLC-8024 cells with low endogenous GDF1 expression with LSD1 inhibitors, and found that inhibition of LSD1 could increase the expression of representative CTA, the GAGE12E, in a dose dependent manner (Fig. S5D). To further confirm whether GDF1 is required for the increase of GAGE12E by LSD1 inhibitor, CRISPR-cas9 mediated knock out of GDF1 was performed in Hep3B cells with high GDF1 expression. As shown in Fig. S5E, LSD1 inhibitor can increase the expression of GAGE12E even in the absence of GDF1 expression. These findings indicated that GDF1 is not indispensable for LSD1 inhibitor in increasing GAGE12E expression. Inhibiting LSD1 in GDF1 negative tumors might also be able to increase tumor immunogenicity. However, the extent of GAGE12E upregulation in GDF1 null Hep3B cells was much weaker compared with that in wildtype Hep3B cells, when treating the same doze of LSD1 inhibitor. Similar results were also found in PLC-8024 cells. The extent of GAGE12E upregulation in GDF1-transfected PLC-8024 cells was much higher compared with that in wildtype PLC-8024 cells (Fig. 5H). These evidences indicated that the effect of LSD1 inhibitor in inducing CTAs expression was boosted in the presence of GDF1. Although the detailed mechanism is still unclear, we believe that the presence of GDF1 might induce specific cofactors to prime the tumor microenvironment, and favor LSD1 in regulating tumor immunogenicity. Based on our current evidences, we speculate that GDF1 might

serve as a potential indicator for LSD1 inhibitors in activation of CTAs in HCC. But we still need more clinical evidences to support the conclusion. To avoid misunderstanding, we have changed the statements in the revised manuscript without specifying “in GDF1 positive HCC tumors”. Relative changes have been made and new data have been incorporated into the “Abstract” section, the “Results” section, (2nd paragraph of page 11, 1st paragraph of page 12), the “Discussion” section (1st paragraph of page 19), as well as in Fig. 5H, Fig. S5, D, E.

5. It is unclear how GDF1 represses LSD1 expression? Can SMAD3 directly be recruited to LSD1 promoter and repress transcription? This reviewer is also intrigued regarding the difference between GDF1 and TGF-beta. Does TGF-beta (that also signals via SMAD3) not repress LSD1 expression?>

Our reply:

In accordance with reviewer’s suggestions, we further investigated the molecular mechanism of GDF1 in repressing LSD1 expression, and the role of TGF-beta in this process. Both public database and our ChIP-PCR assay demonstrated that SMAD3 can directly bind to the promoter region of LSD1 (Fig. S5F, Fig. 5I). To further investigate the difference between GDF1 and TGF-beta in activating SMAD3 and downstream signaling, luciferase reporter assay was performed to test the transcriptional activity of LSD1 promoter in HCC cells treated with GDF1 or TGF-beta respectively. Surprisingly, we found the two cytokines from the same superfamily have totally opposite effects on LSD1 transcription. Treatment with GDF1 suppressed LSD1 transcriptional activity which was in accordance with our previous findings. However, treatment with TGF-beta was found to activate LSD1 transcription in our model (Fig. 5J). To further confirm the regulation of TGF-beta on LSD1 and CTAs, PLC-8024 cells were treated with recombinant TGF-beta. The results showed that TGF-beta also up-regulated LSD1 and suppressed CTAs expression (Fig. S5G). Although SMAD2/3 can be phosphorylated by both GDF1 and TGF- β 1 treatment, the absolute converse downstream effects indicated that SMAD2/3 may recruit different cofactors to regulate its target genes depending on different environmental situations. To further confirm whether GDF1 is required for SMAD2/3-induced LSD1 inhibition, ectopic expression of SMAD3 was induced in GDF1 knock out cell lines. As shown in the results, overexpression of SMAD3 failed to inhibit LSD1 expression and activate GAGE12E when GDF1 is depleted (Fig. 5K). In addition, specific inhibition of SMAD2 or SMAD3 abolished the regulation of GDF1 on both LSD1 and GAGE12E expression (Fig. S5, H and I). These findings indicated that the components of the GDF1/SMAD2/3/LSD1 axis is mutually dependent, and might function only in specified tumor microenvironments. This part of data has been incorporated into the “Results” section, (2nd paragraph of page 12, 1st paragraph of page 13), as well as in Fig. 5, I, J, K, and Fig. S5, F, G, H, I, and fully discussed in the “Discussion” section (1st paragraph of page 18).

6. Include molecular weight markers on all the Western blots.

Our reply:

In accordance with reviewer's suggestions, we have added molecular weight markers on all the Western blots.

Reviewer #2 (Remarks to the Author):

In their manuscript, Wei Cheng et al. suggest that GDF1-induced tumor plasticity drives cancer-testis antigen presentation, with implications for GDF1 expression as a biomarker for extending the therapeutic window for immunotherapy. I was asked to provide my expert opinion on the technical aspects of RNA sequencing in this study, and notified that all other areas are covered by alternate reviewers. I am therefore limiting my response solely to this facet of the study.

RNA sequencing was performed on PLC-8024 cells transfected with GDF1 or control vectors. This was used to explore differentially expressed genes, illustrated in Figure 5A and S3A and briefly described in-text. I reviewed the full description of the technical aspects of the RNA sequencing experiment from this work, and will reproduce it here:

“RNA sequencing was performed in Shanghai Shengong biotech co. ltd.”

This information is insufficient for me to gauge the rigor and suitability of the experiment in supporting the results described in this study. There is no data availability statement, and no data (including summarized expression values) from the RNA sequencing experiment described in this work (notably there is the Table S1 dataset provided for the related rtPCR experiments).

For me to properly evaluate the technical merit of this aspect of the work, the additional information should be provided:

- Library preparation description and sequencing technology
- QC metrics, including at a minimum RIN values, alignment quality distribution, percentage uniquely mapped reads, 5'/3' bias distribution, and insert size distribution (if applicable).
- Read alignment and expression estimation software used, and associated parameters
- Expression summary files

Our reply:

In accordance with reviewer's suggestions, we have described the library preparation and sequencing methodology in detail in the “Materials and Methods” section (2nd paragraph of page 22). In addition, the QC metrics, including RNA quality control, alignment quality distribution, percentage uniquely mapped reads, 5'/3' bias distribution, and insert size distribution et al. have been shown in Fig. S9. Read alignment and expression estimation software used, and associated parameters were also shown in the “Materials and Methods” section (1st paragraph of page 23). Expression summary files were attached as supplementary documents Tab. S7.

Access to the raw data from the following link:

https://drive.google.com/drive/folders/1Vhdc0jFVRnREo5CL0SUuu1f_ygjUbpAL?usp=sharing

(G0, 8024-CTR; G1, 8024-GDF1)

It is also worth noting that the above description of the RNA sequencing experiments preceded the following:

“DAVID bioinformatics resources (<https://david.ncifcrf.gov/home.jsp>) was used for gene ontology analysis.”

This is likely the foundation for the following statement from the results:

“the most differentially expressed genes were enriched in the cancer testis antigens, which include the GAGE family members”

However, without further description of the analyses undertaken to generate this result, the subset of GAGE family gene symbols overlaid on panel 5A and S3A appear cherry-picked to support the narrative of this work. Many members of the GAGE family—and other CTAs—are not mentioned or described.

A formal gene set enrichment analysis of all CTAs or GAGE-family CTAs (with gene family members as defined by publicly-maintained and recognized datasets, such as those at <http://www.cta.lncc.br/modelo.php>) might go a long way towards enforcing this stated result from the RNA sequencing data.

Our reply:

Thanks for the reviewer’s suggestions. Actually, the CTAs aroused our great interest because we found many GAGE and MAGE family members appeared in the list of the most up-regulated genes induced by GDF1 overexpression. We acknowledge that it is not proper to make any conclusions at this stage without stringent statistical analysis. To confirm the association between GDF1 and CTAs, a formal gene set enrichment analysis of all CTAs including GAGE-family CTAs, defined by publicly-maintained and recognized datasets was analyzed. As shown in Fig. 5A, significant enrichment of CTAs including GAGE-family members was found in GDF1 downstream targets. To avoid misunderstanding, the gene expression volcano figure was removed and replaced with the GSEA results. Relevant changes of the statement have also been made in the “Results” section (2nd paragraph of page 10). Actually, in our experimental validation part, we confirmed the up-regulation of a broad panel of CTAs (including GAGEs, MAGEs, and LY6 family members) activated by GDF1 overexpression. We hope the revised manuscript can satisfy the reviewer’s concerns and requirements.

Reviewer #3 (Remarks to the Author):

This manuscript is potentially of great importance. For it suggests new ways to identify human liver cancer patients (from the serum biomarker, GDF1) amenable to specific molecular immunotherapy treatments that might augment survival significantly.

The major claims of this paper are novel and, without question, will be interesting to others in the hepatology community as well as the wider fields of human cancer and immunotherapy.

However, the work and its conclusions, for the most part convincing (the major primary observations are made with tissue biopsies from human patients suffering from hepatocellular carcinoma [HCC]), is based upon evidence of association.

It is suggested therefore that the work would be strengthened by inclusion of:

1. Immunohistochemical staining of fetal mouse livers (Fig. 3A) and by improved images of GDF1 immunohistochemical staining at negative, low, moderate and high staining intensities in HCC (Fig. S1, C; the latter images appear to have been made from tissue sections subjected to fixation artifact).

Our reply:

In accordance with reviewer's suggestions, we have checked the staining of GDF1 in fetal mouse livers at different embryonic days. As shown in Fig. S2A, the IHC staining showed similar trend with the qPCR results. High staining of GDF1 was found in early embryos and e16.5 embryonic livers, but disappeared in postnatal livers. This result further confirmed the activation of GDF1 in embryonic liver development. Improved images of GDF1 immunohistochemical staining at negative, low, moderate and high staining intensities in HCC were replaced in the revised version of the manuscript. The new data was incorporated into the "Results" section (2nd paragraph of page 7), as well as in Fig. S1C, S2A.

2. Partial hepatectomy and primary hepatocyte culture studies with wild type mice (or rats). Retro-differentiation phenomena occur in both models, so GDF1 expression is expected. The herculean work effort reported here relied upon HCC cell lines, not primary HCC cells. For example, will GDF1 block proliferation of normal hepatocytes?

Our reply:

In accordance with reviewer's suggestions, we have further investigated the role of GDF1 in a partial hepatectomy mice model, and primary tissue organoids culture model. As shown in Fig. S2B, positive staining of GDF1 was found in the mice liver 3-5 days after partial hepatectomy, but disappeared after liver regeneration. This finding further confirmed the importance of GDF1 in liver development and regeneration. To investigate how GDF1 affects primary normal hepatocytes proliferation, a 3D organoids system was established to culture primary liver tissues derived from six-week-old C57BL/6 mice. Lentivirus-mediated transfection of mGDF1 or control vector was performed in the organoids, and Ki67 immuno-fluorescent staining was performed to monitor cell proliferation. As shown in Fig. S2C, overexpression of mGDF1 significantly inhibited primary hepatocytes proliferation reflected by Ki67 staining. This was in accordance with the findings in HCC cell lines, that GDF1 could suppress cell proliferation. These parts of data have been incorporated into the "Results" section (2nd paragraph of page 7, 1st paragraph of page 8), as well as in Fig. S2, B and C.

3. DEN-chemical hepatocarcinogenesis and immunotherapy studies with wild type (control) and hepatocyte-specific GDF1 knockout mice would greatly strengthen and prove the paper's major conclusions without doubt.

Our reply:

Thanks for the reviewer's suggestions. The DEN-chemical hepatocarcinogenesis mice model is an ideal model in investigating the in vivo function of specified genes during liver carcinogenesis. Actually, we have considered to establish a DEN mice model to investigate the role of GDF1 in HCC immunotherapy. However, there are two problems restricted us from utilizing this model. The first problem is that DEN-induced mice liver tumor does not express GDF1. We have checked the expression of GDF1 in publicly available databases of previously established DEN-induced mice or rat liver cancer models (GSE141090 rat; GSE93392, mouse). The results showed that the expression of GDF1 was not elevated after DNE treatment (Chen et al. *iScience* 2020; 23:101690). GDF1 is not expressed in normal liver tissues, and it was only activated in a proportion of HCC tumor tissues in the clinic. Considering that the liver carcinogenesis is a complex process which might be induced by multiple molecular disorders, activation of GDF1 cannot account for all types of liver carcinogenesis. Unfortunately, the liver tumors induced in the DEN-chemical hepatocarcinogenesis mice model might not be able to represent GDF1 positive liver tumors. In this case, it would be difficult to observe phenotypic changes in DEN-induced liver tumors when GDF1 was knocked out. The second problem is that the DEN-induced liver tumors have been reported to have low immunogenicity (Yim et al. *Mol Cancer Res* 2018; 16:1713-1723). Knock out of GDF1 is supposed to further decrease tumor immunogenicity. It would be difficult to evaluate immunotherapeutic responses in tumors with low immunogenicity. Considering the two problems, we didn't use the DEN-chemical hepatocarcinogenesis mice model. Instead, a GDF1 overexpression xenograft mice model was used in our

study.

Considering the reviewer's concern of down-regulating GDF1 in liver carcinogenesis, CSIPER-CAS9 mediated GDF1 specific knock out was performed in Hep1-6 cells (Fig. S8A). The GDF1^{KO} and GDF1^{WT} Hep1-6 cells were intrasplenic injected into C57 mice, and further treated with anti-PD1 therapy. Considering that the tumor immunogenicity is low in both GDF1^{WT} and GDF1^{KO} Hep1-6 cells, LSD1 inhibitors was used to enhance tumor immunogenicity during anti-PD1 treatment. As shown in the results, knock out of GDF1 dramatically reduced the tumor formation and metastasis in mice model (Fig. S8, B, C, D). Only one small liver nodule was found in 1 out of 9 mice when high concentration of cells were injected (3.5×10^6 cells/mice). These findings indicated that GDF1 is critical for HCC tumorigenesis and metastasis. In the control GDF1^{WT} group, combination of LSD1 inhibitor with anti-PD1 antibody showed significant superior therapeutic advantages compared with single treatment groups in both suppressing tumor growth and metastasis, and prolonging overall survival (Fig. 6, H, I and J, Fig. S8, E, F, and G). IHC staining also showed that the tumor infiltrating CD8+/GZMB+ cytotoxic T lymphocytes were greatly increased in the combination group than in the single treatment groups (Fig. S8H). These finding further supported the clinical use of LSD1 inhibitors in combination with immune checkpoint inhibitors in HCC treatment. The new data was incorporated into the "Results" section (2nd paragraph of page 15), as well as in Fig. 6, H, I, J, Fig. S8, A-H, and fully discussed in the "Discussion" section (1st paragraph of page 19).

4. The paper will definitely influence thinking in the field. For example, is it by chance that GDF1 – a protein involved in the determination of left-right symmetry during development – functions via hepatic ALK4/7-SMAD3 signaling cascades? What are the additional mechanisms by which GDF1 reactivates cancer testis antigens in HCCs?

Our reply:

Thanks for the reviewer's positive comments. Actually, we believe it is not by chance that a development protein is critical in tumorigenesis. We have evidences that the oncofetal gene GDF1 is activated in multiple cancer types and significantly associated with patients' poor prognosis (data from publicly available databases). Using shRNA mediated specific knock down of the critical components of the ALK/SMAD signaling cascade, we proved that GDF1 functions through ALK7 receptor, and downstream SMAD2/3 signaling, which was also addressed in the first question of reviewer 1. In addition, we found knock out of GDF1 dramatically reduced the tumorigenicity and metastatic ability of HCC cells in vivo. All these evidences indicated that the reactivation of GDF1 in tumor cells was not simply by chance, but significantly contributed to tumor malignant transformation. We believe there might be multiple mechanisms of GDF1 in activating CTAs in HCC. Considering that GDF1 can activate a broad panel of CTAs, the most potential mechanism is through epigenetic regulation. From our screening data, we found GDF1 can actually

influence series of epigenetic regulators. Among them, LSD1 is the most significantly affected. We do not preclude the possibility that GDF1 can potentially affect other epigenetic regulators. We also have evidences that the histone modification status was changed in the promoter region of multiple CTAs after GDF1 treatment (unpublished data). This is a potential interesting field which need further intensive investigation. This part has been fully discussed in the “Discussion” section of the revised manuscript (1st paragraph of page 19).

5. Statistical analyses are appropriate and valid throughout.

Given the level of detail provided, the experiments should be reproducible.

The manuscript should be carefully rewritten as it is filled throughout with English language errors.

Reviewed by HL Leffert.

Our reply:

In accordance with reviewer’s suggestions, the revised manuscript was carefully edited by a native speaker from a professional authority providing language editing services.

Reviewers' Comments:

Reviewer #1:

Remarks to the Author:

The authors have greatly improved their manuscript. There are a few minor things that need attention.

Specific minor comments:

1. When abbreviations are introduced first in the text they need to be explained. For example, ALK7, LASD1, KRT19, AFP, ARG1, TTR, EPCAM, etc.
2. Molecular weight markers need to be included in Figure 1C.
3. What is CAGEE12E? How does it relate to the story.
4. Figure 4E,F. The authors claim a requirement for both SMAD2 and SMAD3 in the GDF1 response upon the SMAD genetic depletion experiments. This may be true, but in order to make that conclusion the authors need to show that SMAD2 knock down does not affect SMAD3 expression and vice versa. The SMAD2 and SMAD3 cDNA sequences are highly similar.
5. Surprisingly, GDF1 and TGF-beta have opposite effects on LSD1 expression. Authors discuss role of differential recruitment of co-factors. I wonder whether differential non-SMAD signaling pathways that are activated by GDF1 and TGF-beta might contribute to this effect. This could be discussed as an option.
6. Figure 4C. The knockdown of ALK4 (activin type I receptor) strongly potentiates the sphere formation in the absence or presence of GDF1. This might be noteworthy to mention in the results section (as also IHHA, encoding Activin A) is the second hit after GDF1 in the differential expressed genes in Supplementary Figure 1B.
7. In the Methods section, include details on compounds that are used in this study (small molecule inhibitors, antibodies). Also include a section on how the statistical analysis was performed.

Reviewer #3:

Remarks to the Author:

The authors have elegantly responded to all of this Reviewer's principal experimental concerns. The manuscript deserves publication.

Note to Authors: Your reply to my 3rd comment (which dealt with DEN-Induced HCC) raised even more intriguing puzzles to my comment. Perhaps these points could be added as a footnote?

Reviewer #4:

Remarks to the Author:

My role in this review is limited to data analysis. Here are my main comments.

Survival analysis

- a) It is necessary to describe how the two groups (GDF1 high and low) of samples in Figures 1H and 1I were identified. Was the GDF1 protein expression level >2.0 used as the cutoff? If so, how was this cutoff chosen? What would the survival plots look like if the samples were divided into half?
- b) Lines 127-129 and 915-917: the authors should specify how the p-values were computed: log-rank test or the Wilcoxon test.

c) Throughout the manuscript, "log-rank analysis" should be "log-rank test".

d) Overall survival and disease-free survival should be defined.

Gene set enrichment analysis

a) Lines 976-978: the authors stated that "RNA seq was performed in 8024 CTR and 8024 GDF1 cells. Gene set enrichment analysis was performed in PLC 8024 cells transfected with or without GDF1 to examine the enrichment in gene sets of cancer testis antigens". It sounds like that RNA-seq analysis was done on control and GDF1 transfected 8024 cells. However, the GSEA (gene set enrichment analysis) was carried out using data from PLC 8024 cells. This needs to be clarified.

b) The authors should describe the number of differentially expressed genes used in the GSEA analysis. The authors considered differentially expressed genes as those with q-value <0.05 and absolute fold change >1.0 . Isn't the fold change cutoff too lenient? How many differentially expressed genes did the authors identify?

c) The authors used both DEGseq and DEseq for differential gene expression analysis. Most readers would be familiar with DEseq but not DEGseq. It would be helpful to have both tools cited. It would also be helpful to state clearly which tool was used for which data set.

d) In GSEA analysis, the authors provided an external list of genes (CTA gene set from CTDatabase) as their signature data set. It would be helpful to specify the number of CTA genes used in the analysis and whether or not the analysis used a pre-ranked gene list.

e) The GSEA plot (Fig. 5A) appears to indicate enrichment. However, the normalized enrichment score (NES) (NES=1.0) suggested otherwise. An NES score of 1.0 means that the enrichment score over the mean enrichment of random samples of the same gene-set (CTA set, in this case) size is 1. Something appears to be strange.

Proof-reading is needed. For example, line 562 "was used for differently analysis" should be "was used for differential gene expression analysis".

Point-by-point response to comments of Reviewers

Reviewer #1 (Remarks to the Author):

The authors have greatly improved their manuscript. There are a few minor things that need attention.

Specific minor comments:

1. When abbreviations are introduced first in the text they need to be explained. For example, ALK7, LASD1, KRT19, AFP, ARG1, TTR, EPCAM, etc.

Our reply:

In accordance with reviewer's suggestions, the full name of the abbreviations were added when they are introduced first in the text. Relative changes have been made in the "Abstract" section as well as in the main text.

2. Molecular weight markers need to be included in Figure 1C.

Our reply:

In accordance with reviewer's suggestions, molecular weight markers were added in Figure 1C of the revised manuscript.

3. What is CAGEE12E? How does it relate to the story.

Our reply:

GAGE12E is a representative member of the GAGE family, which belongs to cancer testis antigens. The GAGE family members share great similarity with each other, and most of the GAGE12 members were significantly upregulated by GDF1 overexpression in our data. Thus an antibody recognizing GAGE12B/C/D/E was selected to detect the GAGE abundance at protein level. This part has been explained in the "Results" section (2nd paragraph of page 10).

4. Figure 4E,F. The authors claim a requirement for both SMAD2 and SMAD3 in the GDF1 response upon the SMAD genetic depletion experiments. This may be true, but in order to make that conclusion the authors need to show that SMAD2 knock down does not affect SMAD3 expression and vice versa. The SMAD2 and SMAD3 cDNA sequences are highly similar.

Our reply:

The shRNAs were designed specifically targeting SMAD2 or SMAD3 respectively to avoid off-target effect. In accordance with reviewer's suggestions, we have further checked whether there is cross reaction of these shRNAs by western blot analysis. As shown in Fig4E-F and Fig5H-I, shRNA-mediated genetic depletion of SMAD2 cannot inhibit SMAD3 abundance. Meanwhile, genetic depletion of SMAD3 also cannot inhibit SMAD2 abundance. This part of data have been incorporated into the relative figures of the revised manuscript.

5. Surprisingly, GDF1 and TGF-beta have opposite effects on LSD1 expression.

Authors discuss role of differential recruitment of co-factors. I wonder whether differential non-SMAD signaling pathways that are activated by GDF1 and TGF-beta might contribute to this effect. This could be discussed as an option.

Our reply:

Thanks for the reviewer's valuable comments, we were also surprised that GDF1 and TGF-beta have opposite effects on LSD1 expression. Actually the TGF- β -SMAD signaling pathway is very complicated. From literature review, we found that the activated SMAD2/3 might recruit different cofactors to either activate or repress downstream gene expression. It is also possible that GDF1 and TGF-beta might activate different non-SMAD signaling pathways to regulate LSD1 expression. These need further detailed investigation in our future work. In accordance with reviewer's suggestions, this phenomenon and possible mechanisms have been substantially discussed in the "Discussion" section of the revised manuscript (1st paragraph of page 18).

6. Figure 4C. The knockdown of ALK4 (activin type I receptor) strongly potentiates the sphere formation in the absence or presence of GDF1. This might be noteworthy to mention in the results section (as also IHHA, encoding Activin A) is the second hit after GDF1 in the differential expressed genes in Supplementary Figure 1B.

Our reply:

Thanks for the reviewer's valuable comments and suggestions. The second hit after GDF1 in the differential expressed genes also functions through the ALK receptor is an important clue to indicate that the ALK receptors might be important to mediate extracellular signaling in maintaining HCC tumor lineage plasticity. This part has been incorporated into the "Results" section in accordance with reviewer's suggestions (1st paragraph of page 10).

7. In the Methods section, include details on compounds that are used in this study (small molecule inhibitors, antibodies). Also include a section on how the statistical analysis was performed.

Our reply:

In accordance with reviewer's suggestions, details on compounds that are used in this study were added in the "Materials and Methods" section (1st paragraph of page 28). Methods for the statistical analysis were also added in the "Materials and Methods" section (2nd paragraph of page 30).

Reviewer #3 (Remarks to the Author):

The authors have elegantly responded to all of this Reviewer's principal experimental concerns. The manuscript deserves publication.

Note to Authors: Your reply to my 3rd comment (which dealt with DEN-Induced HCC) raised even more intriguing puzzles to my comment. Perhaps these points could be added as a footnote?

Our reply:

Thanks for the reviewer's valuable comments and suggestions. We have fully discussed the chosen of animal models regarding the 3rd comment. This part has been incorporated into the "Discussion" section in the revised manuscript (1st paragraph of page 20).

Reviewer #4 (Remarks to the Author):

My role in this review is limited to data analysis. Here are my main comments.

Survival analysis

a) It is necessary to describe how the two groups (GDF1 high and low) of samples in Figures 1H and 1I were identified. Was the GDF1 protein expression level >2.0 used as the cutoff? If so, how was this cutoff chosen? What would the survival plots look like if the samples were divided into half?

Our reply:

IHC slides were scanned using an Aperio CS2 Digital Pathology Scanner (Leica, Germany) at $\times 20$ magnification. Positive staining was assessed using a 5-point scoring system: 0 (0 positive cells), 1 (<10 positive cells), 2 (10–35% positive cells), 3 (36–70% positive cells), and 4 ($>70\%$ positive cells). Meanwhile, the intensity of positive staining was also assessed: 0 (negative), 1 (weak), 2 (moderate), and 3 (strong). The expression of target protein index was calculated as follows: expression index = (positive score) \times (intensity score). Optimal cut-off values for this score system were identified as follows: high expression was defined as an expression index score of ≥ 4 , and low expression was defined as an expression index score of <4 . Detailed description of patient classification was incorporated into the "Material and Methods" section of the revised manuscript (1st paragraph of page 27).

If the samples were divided into half, the plot would look like as below:

b) Lines 127-129 and 915-917: the authors should specify how the p-values were computed: log-rank test or the Wilcoxon test.

Our reply:

In accordance with reviewer’s suggestions, the statistical methods have been incorporated into relative sections of the revised manuscript. The p-values in lines 127-129 and 915-917 were computed using log-rank test.

c) Throughout the manuscript, “log-rank analysis” should be “log-rank test”.

Our reply:

In accordance with reviewer’s suggestions, relative changes have been made in the revised manuscript.

d) Overall survival and disease-free survival should be defined.

Our reply:

In accordance with reviewer’s suggestions, the median overall survival time was added in the “Results” section (1st paragraph of page 6).

Gene set enrichment analysis

a) Lines 976-978: the authors stated that “RNA seq was performed in 8024 CTR and 8024 GDF1 cells. Gene set enrichment analysis was performed in PLC 8024 cells transfected with or without GDF1 to examine the enrichment in gene sets of cancer testis antigens”. It sounds like that RNA-seq analysis was done on control and GDF1 transfected 8024 cells. However, the GSEA (gene set enrichment analysis) was carried out using data from PLC 8024 cells. This needs to be clarified.

Our reply:

Actually, 8024 CTR and 8024 GDF1 cells were PLC-8024 cells stably transfected with control vector or GDF1 respectively. “8024 CTR and 8024 GDF1 cells” have all been changed to “PLC-8024-CTR and PLC-8024-GDF1 cells” in the revised manuscript.

b) The authors should describe the number of differentially expressed genes used in the

GSEA analysis. The authors considered differentially expressed genes as those with q-value <0.05 and absolute fold change >1.0 . Isn't the fold change cutoff too lenient? How many differentially expressed genes did the authors identify?

Our reply:

The selection criteria for differentially expressed genes in our study is $\log_2\text{FoldChange} >1.0$, not absolute fold change. We apologize for the typo made in the previous manuscript. Using $\log_2\text{FoldChange} >1.0$ as cutoff value, 1811 differentially expressed genes were identified, including 545 up-regulated genes and 1266 down-regulated genes. This part of data and relative changes have been incorporated into the "Material and Methods" section of the revised manuscript (1st paragraph of page 24).

c) The authors used both DEGseq and DESeq for differential gene expression analysis. Most readers would be familiar with DESeq but not DEGseq. It would be helpful to have both tools cited. It would also be helpful to state clearly which tool was used for which data set.

Our reply:

For the samples without biological repetition, DEGseq (version 1.26.0) was used for differential gene expression analysis. For the samples with biological repetition, DESeq (version 1.12.4) was used for analysis. In this study, only DEGseq (version 1.26.0) was used for differential gene expression analysis. To avoid misunderstanding, we made relative changes in the "Material and Methods" section of the revised manuscript (1st paragraph of page 24).

d) In GSEA analysis, the authors provided an external list of genes (CTA gene set from CTDatabase) as their signature data set. It would be helpful to specify the number of CTA genes used in the analysis and whether or not the analysis used a pre-ranked gene list.

Our reply:

In accordance with reviewer's suggestions, the number of CTA genes used in the analysis was specified in the "Material and Methods" section of the revised manuscript (2nd paragraph of page 24). The full gene list was also added as Tab. S8. The analysis wasn't performed with a pre-ranked gene list.

e) The GSEA plot (Fig. 5A) appears to indicate enrichment. However, the normalized enrichment score (NES) (NES=1.0) suggested otherwise. An NES score of 1.0 means that the enrichment score over the mean enrichment of random samples of the same gene-set (CTA set, in this case) size is 1. Something appears to be strange.

Our reply:

Thanks for the reviewer's reminding, we have carefully examined our GSEA analysis, especially for parameters setting. We found that we selected "Phenotype" for "Permutation type" in the previous analysis. According to the official GSEA user guide, "Phenotype" is recommended when there are at least 7 samples in each phenotype. When there are fewer samples (less than 7), the parameter "Gene set" is recommended

for “Permutation type”. Considering that our sample size is less than 7, “Gene set” was used as the setting of “Permutation type” to reanalyze the data. The adjusted NES value is 1.407. Relative changes have been made in Fig. 5A, and the detailed parameter setting for GSEA analysis was added in the “Material and Methods” section of the revised manuscript (2nd paragraph of page 24).

Proof-reading is needed. For example, line 562 “was used for differently analysis” should be “was used for differential gene expression analysis”.

Our reply:

In accordance with reviewer’s suggestions, the revised manuscript has been proof-red by a native speaker. Grammar errors have been corrected.

Reviewers' Comments:

Reviewer #1:

Remarks to the Author:

The authors have further improved their manuscript.

A few typos and small issue remaining:

Page 2.

ALK7 abbreviation is wrongly explained. Anaplastic lymphoma kinase 7 should be replaced by Activin receptor-like kinase 7

Page 10.

I suggest change the following text (and perhaps better move that to discussion section):
Interestingly, we noticed that Inhibin alpha (INHA), which ranked the second hit after GDF1 in the differential expressed genes of poorly differentiated HCCs, also functions through ALK receptors. This indicated that the ALK receptors might be important to mediate extracellular signalling in maintaining HCC tumor lineage plasticity

Into the following text (that needs to be adjusted to make a good flow with the remainder of the text):

Of note, Inhibin alpha (INHA), encoding activin A, which ranked the second hit after GDF1 in the differential expressed genes of poorly differentiated HCCs, signals via ALK4 and SMAD2/3. We found that knockdown of ALK4 strongly potentiated the sphere formation in the absence or presence of GDF1 (Figure 4C). It is possible that activin A has a role in maintaining HCC tumor lineage plasticity, and this will be subject of future studies.

Page 19 and 28

Change TGF-beta into TGF- β (greek beta)

Reviewer #4:

Remarks to the Author:

The authors have adequately addressed my comments.

Point-by-point response:

Reviewer #1 (Remarks to the Author):

The authors have further improved their manuscript.

A few typos and small issue remaining:

Page 2.

ALK7 abbreviation is wrongly explained. Anaplastic lymphoma kinase 7 should be replaced by Activin receptor-like kinase 7

Our reply:

Thanks for the reviewer's reminding, we have corrected the abbreviation of ALK7.

Page 10.

I suggest change the following text (and perhaps better move that to discussion section):

Interestingly, we noticed that Inhibin alpha (INHA), which ranked the second hit after GDF1 in the differential expressed genes of poorly differentiated HCCs, also functions through ALK receptors. This indicated that the ALK receptors might be important to mediate extracellular signalling in maintaining HCC tumor lineage plasticity

Into the following text (that needs to be adjusted to make a good flow with the remainder of the text):

Of note, Inhibin alpha (INHA), encoding activin A, which ranked the second hit after GDF1 in the differential expressed genes of poorly differentiated HCCs, signals via ALK4 and SMAD2/3. We found that knockdown of ALK4 strongly potentiated the sphere formation in the absence or presence of GDF1 (Figure 4C). It is possible that activin A has a role in maintaining HCC tumor lineage plasticity, and this will be subject of future studies.

Our reply:

Thanks for the reviewer's kind suggestion and rewriting the paragraph for us. We have revised the manuscript according to the reviewer's suggestion. Relative changes have been made in the "Discussion" section (1st paragraph of page 18).

Page 19 and 28

Change TGF-beta into TGF- β (greek beta)

Our reply:

Thanks for the reviewer's reminding, we have made corrections in the revised manuscript.